# Rhythmic sampling and competition of target and distractor in a motion detection task

**Changhao Xiong[1†], Nathan M Petro[2†], Ke Bo[3], Lihan Cui[1], Andreas Keil[4]\*, Mingzhou Ding[1]\***

[1]J. Crayton Pruitt Family Department of Biomedical Engineering, University of Florida, Gainesville, United States; [2]Institute for Human Neuroscience, Boys Town National Research Hospital, Boys Town, United States; [3]Department of Psychological and Brain Science, Dartmouth College, Hanover, United States; [4]Department of Psychology, University of Florida, Gainesville, United States

## eLife Assessment

This work presents **important** information on rhythmicity of overlapping target and distractor processing and how this affects behaviour. The methods are, in general, clearly laid out and defensible, with several supplementary analyses leading to a **solid** base of evidence for their claims.

**\*For correspondence:**
akeil@ufl.edu (AK);
mding@bme.ufl.edu (MD)

[†]These authors contributed equally to this work

**Competing interest:** The authors declare that no competing interests exist.

**Abstract** It has been suggested that the visual system samples attended information rhythmically. Does rhythmic sampling also apply to distracting information? How do attended information and distracting information compete temporally for neural representations? We recorded electroencephalography (EEG) from participants who detected instances of coherent motion in a random-dot kinematogram (RDK; the target), overlayed on different categories (pleasant, neutral, and unpleasant) of affective images from the International Affective Picture System (IAPS) (the distractor). The moving dots were flickered at 4.29 Hz, whereas the IAPS pictures were flickered at 6 Hz. The time course of EEG spectral power at 4.29 Hz was taken to index the temporal dynamics of target processing. The spatial pattern of the EEG spectral power at 6 Hz was similarly extracted and subjected to a moving-window MVPA decoding analysis to index the temporal dynamics of processing pleasant, neutral, or unpleasant distractor pictures. We found that (1) both target processing and distractor processing exhibited rhythmicity at ~1 Hz and (2) the phase difference between the two rhythmic time courses was related to task performance, i.e., relative phase closer to π predicted a higher rate of coherent motion detection whereas relative phase closer to 0 predicted a lower rate of coherent motion detection. These results suggest that (1) in a target-distractor scenario, both attended and distracting information were sampled rhythmically and (2) the more target sampling and distractor sampling were separated in time within a sampling cycle, the less distraction effects were observed, both at the neural and the behavioral level.

## Introduction

Sustained visual attention is required in many real-life situations such as driving a vehicle or operating machinery and is characterized by limited capacity; not all information available to the visual system can be processed in-depth. Recent work has suggested that to manage the limited capacity problem, the visual system samples the attended information in a rhythmic fashion, mediated by low-frequency intrinsic brain oscillations (*Chota et al., 2022*; *Dugué et al., 2015*; *Fiebelkorn et al.,*

*2013*; *Fiebelkorn et al., 2018*; *Fiebelkorn and Kastner, 2019*; *Helfrich et al., 2018*; *Michel et al., 2022*; *Re et al., 2019*; *VanRullen, 2013*; *Zalta et al., 2020*). In this view, the cycle of a low-frequency intrinsic brain oscillation can be divided into two phases: a high excitability phase and a low excitability phase. When a stimulus occurs during the high excitability phase, behavioral performance tends to be better than average; conversely, if the stimulus occurs during the low excitability phase, performance is generally worse than average (*Lakatos et al., 2008*; *VanRullen, 2013*). Behavioral performance may thus exhibit rhythmic fluctuations at the frequency of the aforementioned low-frequency intrinsic brain oscillation. One paradigm that has been used to test the idea of rhythmic visual sampling is the cue-target paradigm (*Posner, 1980*; *Posner et al., 1987*; *Posner et al., 1988*). The cue at the beginning of each trial, in addition to providing instructions on how the impending target stimulus should be responded to, helps to reset the phase of the low-frequency intrinsic oscillation such that all the trials start at approximately the same phase. By varying the stimulus onset asynchrony (SOA) between the cue and the target, one obtains the behavioral response (e.g. accuracy and/or reaction time) as a function of the SOA. The rhythmic nature and the frequency of this function can then be assessed by applying time-domain and/or spectral-domain analysis.

When attending to one object in isolation, the frequency of rhythmic sampling tends to be in the high theta or low alpha frequency range, i.e., around 8 Hz (*Fiebelkorn et al., 2013*; *Senoussi et al., 2019*; *van der Werf et al., 2023*). When attention is directed to multiple objects in the environment, it has been suggested that rather than sampling all the objects simultaneously, the brain samples the objects in a serial fashion (*Cohen et al., 1990*; *Wyart et al., 2012*). This would then lead to a slower rhythmic sampling of any given object, in the low range of the theta frequency band, i.e., around 4 Hz (*Thigpen et al., 2019*). For example, when participants were cued to attend one visual hemifield but were asked to detect the appearance of a weak stimulus in either the cued or the uncued visual hemifield, the rhythmic detection rate for the target appearing in a given visual hemifield decreased from 8 Hz to 4 Hz (*Chota et al., 2022*; *Fiebelkorn et al., 2013*; *VanRullen, 2013*). Interestingly, when the detection rate functions of the cued and uncued targets were compared, a 180-degree relative phase was apparent, suggesting that the visual system indeed sampled the two visual hemifields in a serial, alternating fashion (*Fiebelkorn et al., 2013*; *Jiang et al., 2024*). In another example, two spatially overlapping clouds of moving dots, one in red color and the other in blue color, moved in orthogonal directions (*Re et al., 2019*), and the participant was cued to attend both the red dots and the blue dots and instructed to report the change in either the red dots or the blue dots as soon as it occurred. When there was only one cloud of moving dots, the detection accuracy exhibited rhythmic fluctuations as a function of the SOA at a frequency around 8 Hz. When both clouds of moving dots were present, rhythmic fluctuations in the accuracy of detecting changes in a given cloud of moving dots were again identified, and the sampling frequency was reduced to 4 Hz. In this case, however, no apparent 180-degree relative phase between the rhythmic behavioral response functions to the red and blue dots was found, suggesting that there was no serial, alternating sampling between the two attended objects if they appeared at the same spatial location.

The real world visual environment contains both task-relevant information (target) and task-irrelevant (distractor) information. It is well established that in the presence of a distractor, the processing of the target is negatively impacted, leading to reduced task performance (*Lavie, 2005*; *Murphy et al., 2016*). This implies that the distractor, despite the need for it to be suppressed by the brain's executive control system (*Kastner et al., 1998*; *Kastner et al., 1999*; *Kastner and Pinsk, 2004*; *Seidl et al., 2012*; *Kastner and Ungerleider, 2000*), is nevertheless processed in the brain, and the competition between the target and the distractor at the neural representational level causes the detriment in behavioral performance. Does the rhythmic sampling theory extend to the target-distractor scenario? If so, what is the temporal relationship between the rhythmic sampling of attended vs distracting stimuli? These questions have hitherto not been addressed. Part of the reason is that the majority of the studies on rhythmic environmental sampling focuses on behavioral evidence, e.g., rhythmicity in the aforementioned performance-vs-SOA function (*Fiebelkorn and Kastner, 2019*; *Landau and Fries, 2012*). Since the distractor is not responded to, its sampling by the visual system cannot be inferred purely on the basis of response behavior, and consequently, it is also not possible to study how the target and the distractor might compete for neural representations purely behaviorally.

In this study, we addressed these limitations by recording neural activities and investigating rhythmic sampling during a target-distractor scenario using steady-state visual evoked potential

(SSVEP) frequency tagging. The stimuli were a cloud of randomly moving dots (the target) super-imposed on emotional images from the International Affective Picture System (IAPS; *Lang et al., 1997*) (the distractor). The target and the distractor were flickered at two different frequencies for an extended duration of ~12 s. The participants were asked to focus on the randomly moving dots and report the number of times the dots moved coherently. In this paradigm, the onset of the stimulus array is the event that resets the phase of the putative low-frequency brain oscillation underlying rhythmic sampling, and the time from the stimulus array onset, referred to as time-from-onset (TFO), is analogous to the SOA in the traditional cue-target paradigm. It is worth noting that, although this paradigm has been used extensively in studies of target-distractor competition with electroenceph-alography (EEG) (*Hindi Attar and Müller, 2012*; *Müller et al., 2008*), it has not yet been examined in the context of rhythmic sampling. Aided by frequency tagging, from the EEG data, we extracted neural representations of target and distractor processing separately as a function of TFO. By exam-ining the rhythmicity of these representations as functions of TFO and the phase relationship between these functions, we assessed (1) whether the target and the distractor were sampled rhythmically and (2) how their temporal competition for neural representations impacted behavioral performance.

## Results

EEG data were collected from 27 subjects performing a sustained attention task. The paradigm is shown in *Figure 1A*. The stimulus consisted of a random-dot kinematogram (RDK; the target) over-layed on different categories (pleasant, neutral, and unpleasant) of affective images from the IAPS (the distractor). The moving dots were flickered at 4.29 Hz, whereas the IAPS pictures were flickered at 6 Hz. The participant was instructed to detect brief episodes of coherent motion (0, 1, or 2) and report the number of such episodes at the end of the trial; each trial lasted ~12 s. Depending on the emotional category of the distracting images, the trials were divided into pleasant trials (28), neutral trials (28), and unpleasant trials (28). Each participant performed 84 trials.

### Behavioral analysis

The overall coherent motion detection accuracy was 55.73% ± 2.94%, with that for pleasant, neutral, and unpleasant trials being 55.67% ± 2.76%, 55.03% ± 3.10%, and 56.48% ± 3.61%, respectively. A one-way ANOVA found no significant difference in behavioral performance between the three types of trials ($F_{2, 78} = 0.053$, p=0.949), suggesting that the three types of distractors exerted similar distracting influence on the detection of coherent motion, irrespective of their emotional significance.

### SSVEP analysis at the whole trial level

The grand average SSVEP at Oz and its Fourier spectrum are shown in *Figure 2*. From *Figure 2B*, spectral peaks corresponding to the flicker frequencies of 4.29 Hz (target) and 6 Hz (distractor) are clearly seen. Filtering the SSVEP between 4.29–0.5 Hz and 4.29+0.5 Hz yielded the signal specific to target processing, whereas filtering the SSVEP between 6–0.5 Hz and 6+0.5 Hz yielded the signal specific to distractor processing. Averaging target amplitude and distractor amplitude across all elec-trodes, the 4.29 Hz amplitude was significantly greater than the 6 Hz amplitude (p=$2.6 \times 10^{-4}$); see *Figure 2C*. SSVEP amplitude topographies for target and distractor in *Figure 2D* showed that the strongest response for both frequencies was concentrated in the occipital channels. In *Figure 2E*, we assessed the relationship between SSVEP amplitude and task performance. Across participants, there was no correlation between target SSVEP amplitude and task performance (p=0.7536); see *Figure 2E*, left. The SSVEP amplitude of the distractor has a slight negative correlation with task performance, indicating that the stronger the distractor processing, the worse the performance, but it is not statistically significant (p=0.1896).

### MVPA at the whole trial level

Our previous work has shown that IAPS pictures from different emotion categories evoke distinct spatial patterns in EEG which can be decoded using machine-learning-based MVPA methods (*Bo et al., 2022*). If we were able to decode the emotion categories of the distractor from the spatial patterns of the 6 Hz SSVEP amplitude, the decoding accuracy can then be used to indicate the strength of the distractor representation in the brain, complementing the 6 Hz SSVEP amplitude considered

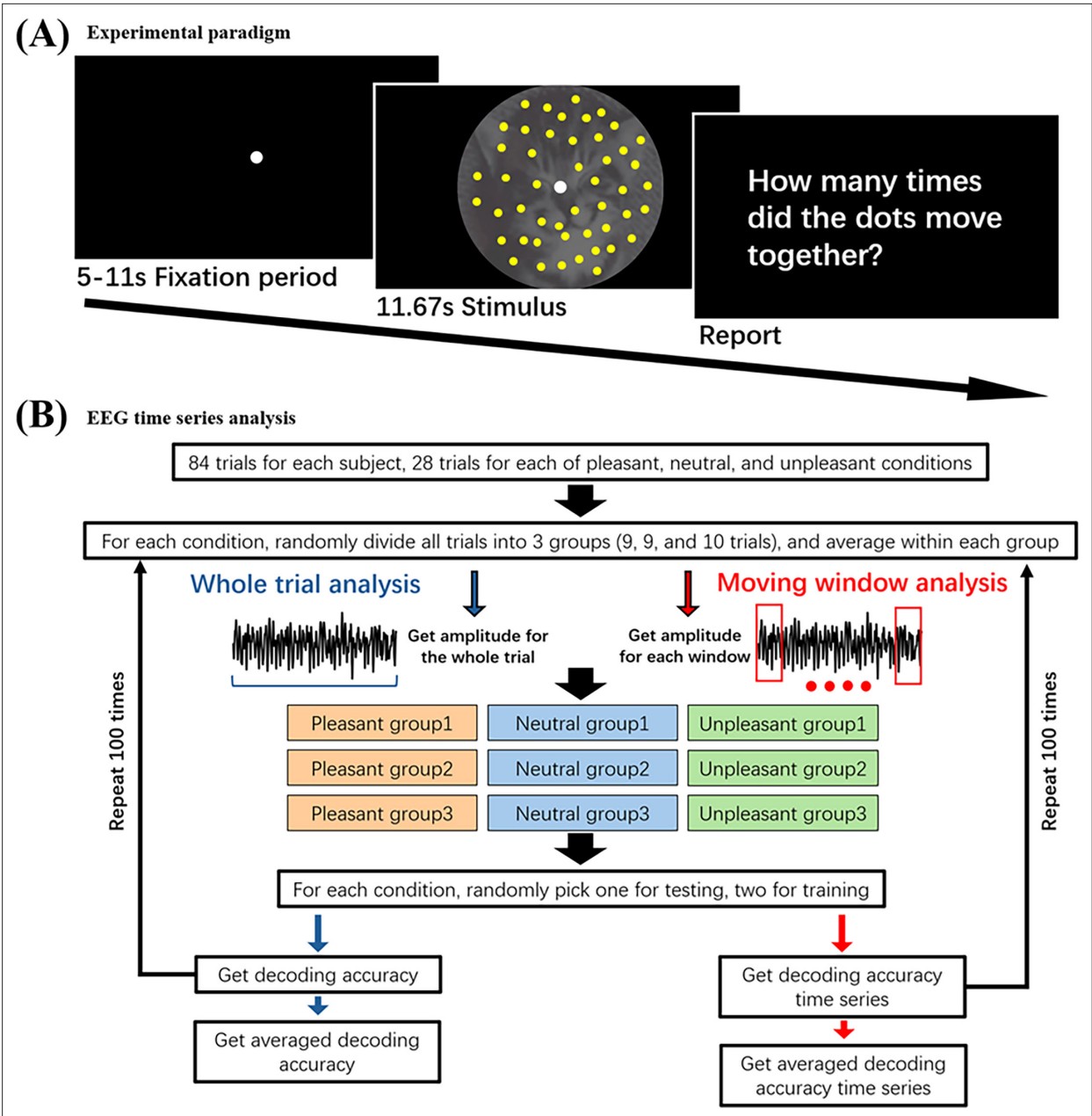

**Figure 1.** Experimental paradigm and general approach for electroencephalography (EEG) data analysis. (**A**) Motion detection task. Randomly moving dots flickered at 4.29 Hz (target) were superimposed in International Affective Picture System (IAPS) images flickered at 6 Hz (distractor). Participants detected brief episodes of coherent motion. (**B**) Target-specific signals and distractor-specific signals were estimated and subjected to (1) whole trial analysis and (2) moving window analysis. MVPA decoding analysis was done using an 'ERP' decoding method. See Materials and methods for more details.

earlier. The decoding was done between different types of emotion trials (e.g. pleasant vs neutral) using an 'ERP' decoding method (*Bae and Luck, 2019*). See *Figure 1B* and Materials and methods for more details. As shown in *Figure 3A*, for pleasant vs neutral, unpleasant vs neutral, and unpleasant vs neutral, the pairwise decoding accuracy was 57.86% ± 9.86%, 55.14% ± 8.17%, and 59.45% ± 9.73%, respectively, which were all significantly above the chance level of 50% at $p=3.2 \times 10^{-4}$, $p=3.0 \times 10^{-3}$, and $p=3.0 \times 10^{-5}$, respectively. As shown in *Figure 3B*, the three-way decoding accuracy was found to be 41.09% ± 6.25%, which is again significantly above chance level of 33.33% at $p=3.9 \times 10^{-7}$. Similar to distractor SSVEP amplitude, no correlation was found between distractor decoding accuracy and task performance; see *Figure 3C*. Also, in order to verify that the distractor decoding accuracy and

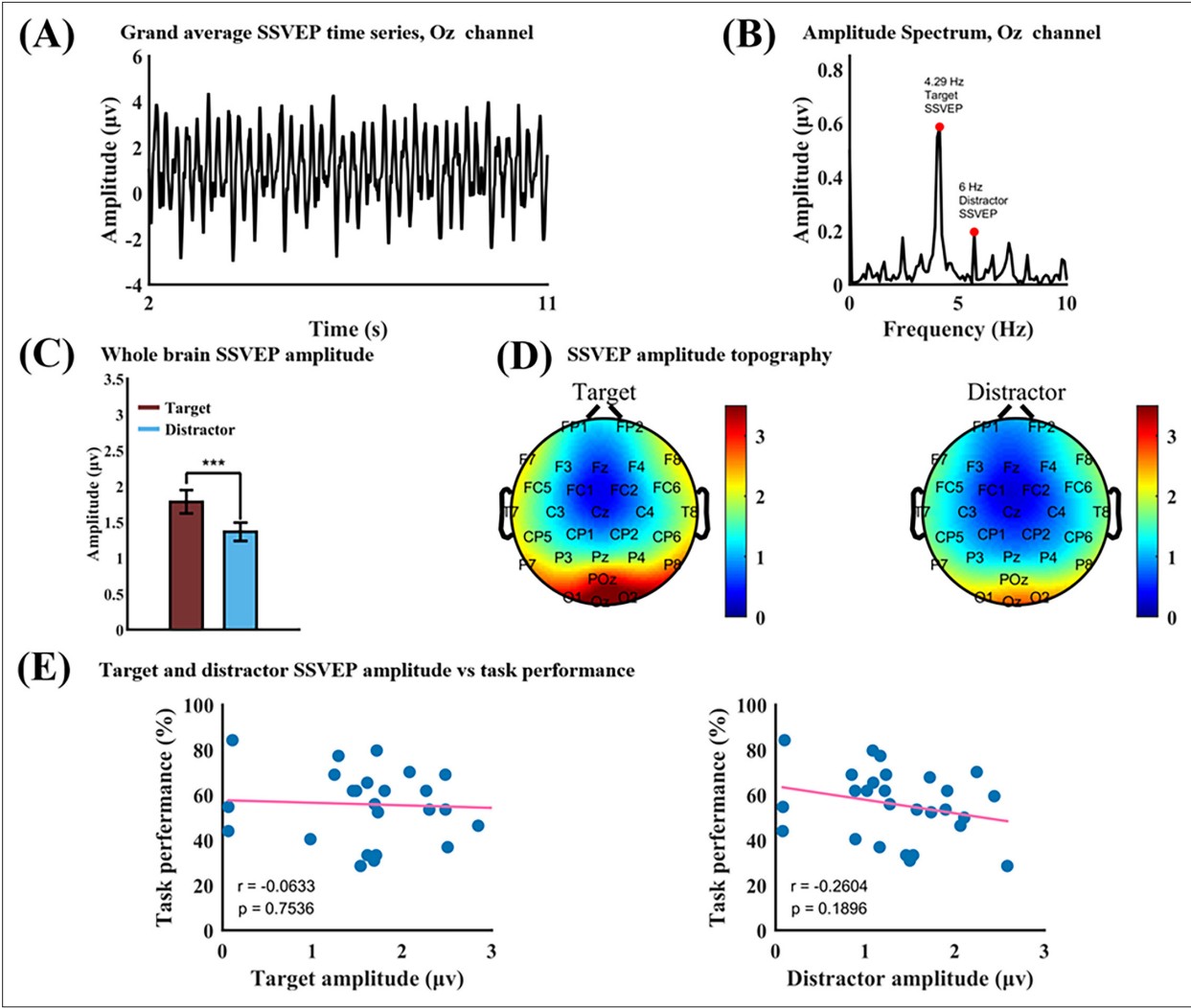

**Figure 2.** Steady-state visual evoked potential (SSVEP) analysis at the whole trial level. (**A**) Grand average SSVEP at Oz. (**B**) Fourier spectrum of the data in A. (**C**) Target amplitude across all electrodes is significantly larger than distractor amplitude at p=2.6 × 10⁻⁴. (**D**) Topographical distributions of target and distractor amplitude. (**E**) Correlation between target SSVEP amplitude and task performance (left) and between distractor SSVEP amplitude and task performance (right). Both correlation values are not significant.

the distractor amplitude were independent indices of distractor processing, we correlated the two across participants. As *Figure 3D* shows, no correlation was found, suggesting that the two quantities provided complementary characterization of distractor processing (also see Appendix 1—figure 3).

## Moving window analysis of target and distractor processing

To examine the temporal dynamics of target processing, the target SSVEP amplitude time series was obtained using the moving window approach, where the window duration was 0.5 s and the step size 0.25 s. Fourier analysis was then applied to assess the rhythmicity of the time series. The results of these analyses for one representative participant are shown in *Figure 4Ai*. The rhythmic nature of target processing is apparent with a spectral peak at ~1 Hz. Across all participants, the averaged Fourier spectrum is shown in *Figure 4Aii*, where the frequency of the spectral peak was found to be 1.08±0.11 Hz. These results supported the idea that the attended target was sampled rhythmically with a sampling frequency at ~1 Hz (delta frequency band).

To examine the temporal dynamics of distractor processing, three-way MVPA decoding was performed for the three types of emotion trials using the moving window approach with the same window duration and step; see *Figure 1A* and Materials and methods for more details. The three-way decoding accuracy time series and the Fourier spectrum from one representative participant are

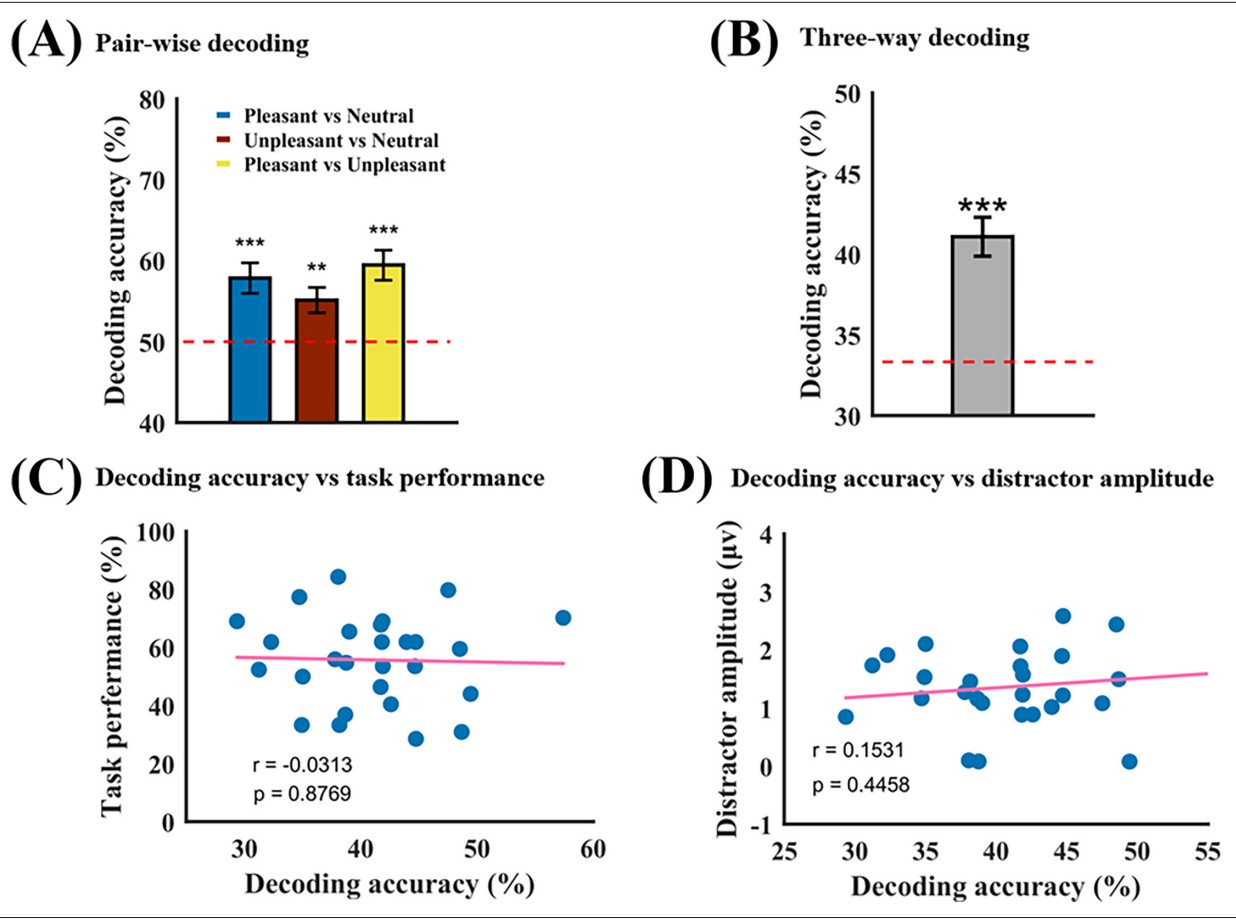

**Figure 3.** MVPA decoding analysis of distractor processing at the whole trial level. (**A**) Pairwise decoding accuracies between pleasant vs neutral, unpleasant vs neutral, and pleasant vs unpleasant are 57.86% ± 9.86%, 55.14% ± 8.17%, and 59.45% ± 9.73%, respectively, which are all significantly above chance level of 50% (red dashed line) at $p=3.2 \times 10^{-4}$, $p=3.0 \times 10^{-3}$, and $p=3.0 \times 10^{-5}$. (**B**) Three-way decoding accuracy is 41.09% ± 6.25%, which is significantly higher than the chance level of 33% (red dashed line) at $p=3.9 \times 10^{-7}$. (**C**) Decoding accuracy vs task performance. The correlation of r=−0.0313 (p=0.8769) is not significant. (**D**) Distractor decoding accuracy vs distractor steady-state visual evoked potential (SSVEP) amplitude. The correlation of r=0.1531 (p=0.4458) is not significant.

shown in *Figure 4Bi*. The rhythmic nature of the decoding accuracy time series is again apparent, and the spectral peak is at ~1 Hz. Across all participants, the averaged spectrum is shown in *Figure 4Bii*, where the peak frequency was determined to be 1.08±0.11 Hz. These results supported the idea that the distractor was also sampled rhythmically with a sampling frequency at ~1 Hz (delta frequency band).

### Target-distractor competition and task performance

As shown above, the present evidence suggests that both the target and the distractor were sampled rhythmically, at ~1 Hz. Since the sampling frequency was approximately the same for the two rhythmic time series, the relative phase between them can then be assessed, which characterizes the temporal relationship between the sampling of target and distractor. *Figure 5A* shows the distribution of the relative phase for all participants (mean relative phase = 0.51 ± 0.31π). A Kolmogorov-Smirnov (K-S) test was applied to the relative phase distribution to see whether it departed from the uniform distribution (*Figure 5B*). A K-S statistic of 0.10 showed that the relative phase distribution is not different from the uniform distribution at p=0.92, suggesting that there was no systematic relative phase between rhythmic samplings of target vs distractor across participants.

Since simultaneously presented target and distractor compete for neural representations and the stronger the competition, the worse the task performance, one may expect that if the target sampling and the distractor sampling are well separated in time, namely, if they occur in opposite phases of

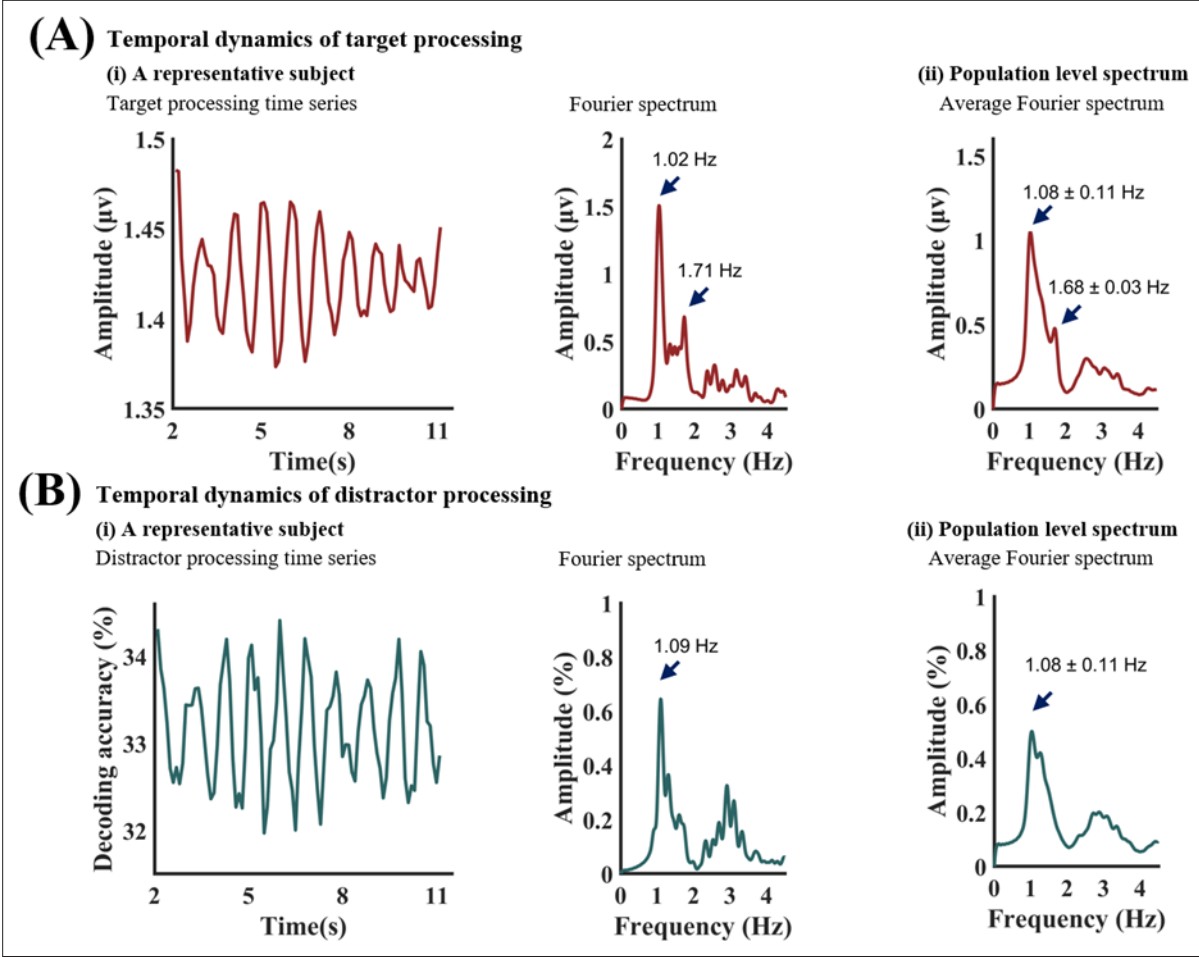

**Figure 4.** Temporal dynamics of target and distractor processing. (**A**) (**i**) Target amplitude time series from the moving window approach for a representative subject (left) and its Fourier spectrum (right). (**A**) (**ii**) The average target amplitude spectrum across 27 subjects. (**B**) (**i**) Distractor decoding accuracy time series from the moving window approach for a representative subject (left) and its Fourier spectrum (right). (**B**) (**ii**) The average distractor decoding accuracy spectrum across 27 subjects.

the 1 Hz brain oscillation, the competition will be minimized, and the task performance will be maximized. Conversely, if the target and the distractor were sampled during the same phase within the 1 Hz cycle, the target-distractor competition will be maximized, and the task performance will be minimized. This notion is tested in *Figure 5C*, using data from a high-performing participant (accuracy = 84.34%) and a low-performing participant (accuracy = 33.33%). Here, the target processing time series and the distractor processing time series were z-scored so they can be displayed in the same graph. In the high performer, the two time courses are highly anticorrelated (relative phase is around $\pi$), indicating that the target and the distractor were sampled in opposite parts of the cycle, while for the low performer, the two time courses are highly correlated (relative phase is around 0), indicating that the target and the distractor were sampled in the same part of the cycle. Across all participants, as shown in *Figure 5D*, a significant positive correlation between relative phase and task performance was observed (r=0.6041, p=0.0008), suggesting that the more the target sampling and the distractor sampling are separated in time (i.e. in opposite phases of the cycle), the less the interference between target and distractor processing, the better the task performance.

## Discussion

In natural vision, task-relevant information (the target) and task-irrelevant information (the distractor) often appear at the same time, and often overlap in visual space. The distractor information, upon entering the nervous system, interferes with the neural representations of task-relevant information,

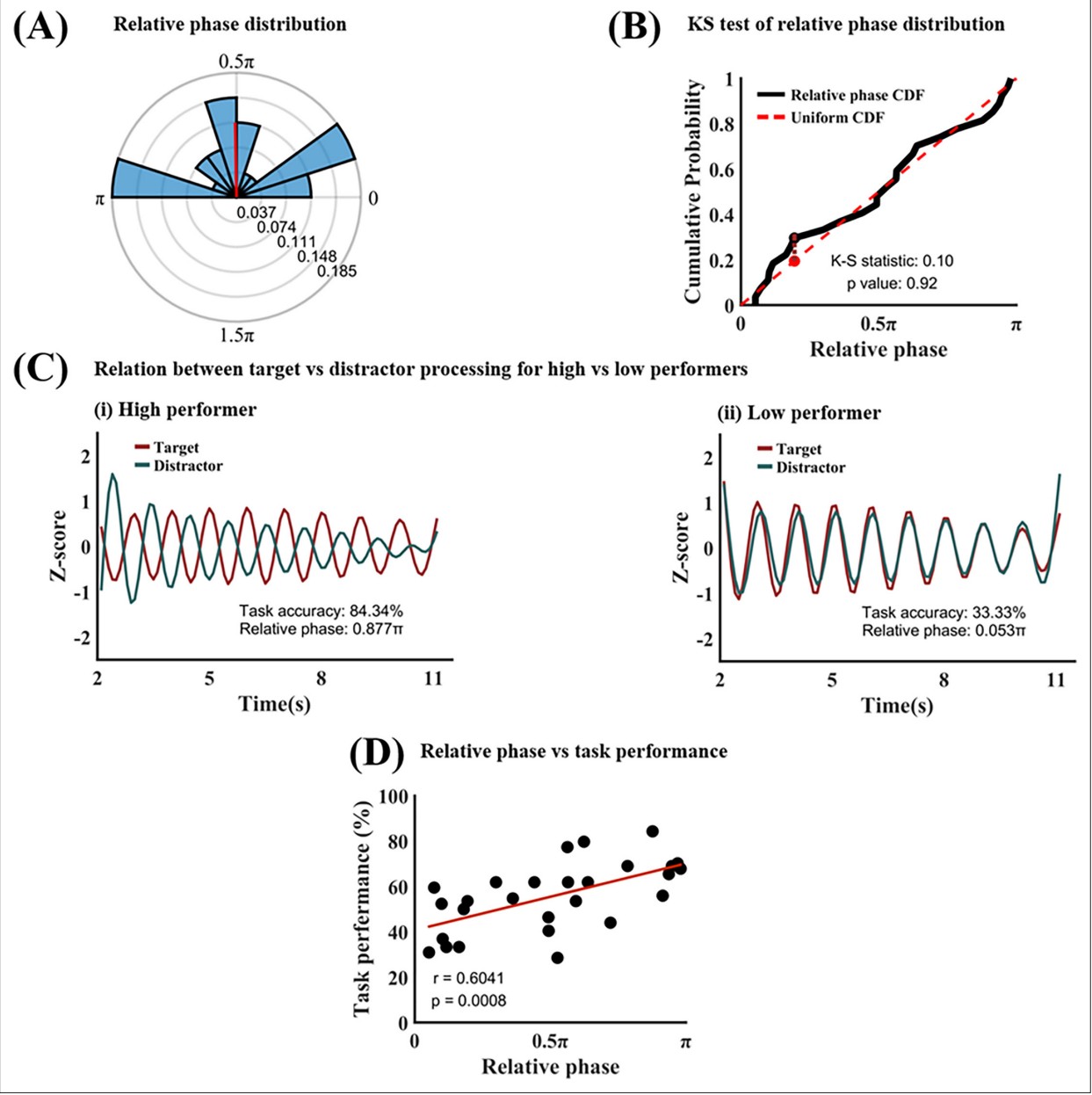

**Figure 5.** Target-distractor competition analysis. (**A**) Phase polar histogram for the relative phase between target processing time series and distractor processing time series (1 Hz). The average relative phase is 0.51π. (**B**) Kolmogorov-Smirnov test showed that the relative phase distribution is not different from uniform distribution. (**C**) Temporal relationship between target processing and distractor processing for (**i**) a high performer (accuracy = 83.84%; relative phase = 0.877π) and (**ii**) a low performer (accuracy = 33.33%; relative phase = 0.053π). (**D**) Task performance vs 1 Hz relative phase. The significant positive correlation (r=0.6041, p=0.0008) indicated that the more separated the target and distractor sampling within the 1 Hz oscillation cycle, the better the behavioral performance. CDF: cumulative distribution function.

causing degraded task performance (*Deweese et al., 2016*). In this study, we examined the temporal dynamics of target and distractor processing during sustained visual attention by analyzing EEG data from an SSVEP paradigm in which random moving dots (target) flickered at one frequency were super-imposed on IAPS pictures (distractor) flickered at another frequency. In particular, we tested whether rhythmic sampling applied to distracting information and how target-distractor competition affected behavior. The results showed that (1) distractor information (i.e. IAPS pictures from different emotion categories) can be decoded from the distributed patterns of scalp EEG, (2) both the target and the distractor are sampled rhythmically with the same sampling frequency of ~1 Hz (delta frequency band), and (3) the more negative (i.e. closer to 180 degrees) the phase relationship between the sampling of

the target and that of the distractor, i.e., the more temporally separated between target sampling and distractor sampling within a sampling cycle, the better the behavioral performance.

## Rhythmic sampling of attended and ignored information

Previous studies have investigated how attended information is temporally sampled using the cue-target paradigm. In particular, if some behavioral measures such as the stimulus detection accuracy or reaction time are found to be a periodic function of the time between the cue and the target, i.e., the stimulus-onset asynchrony or SOA, then it is taken as evidence in support of rhythmic sampling. If there is only one attended target, the frequency of rhythmic sampling tends to fall in the upper end of the theta frequency band (~8 Hz) (*Busch and VanRullen, 2010*; *Huang and Luo, 2020*; *Huang et al., 2015*; *VanRullen, 2013*). When there are more than one attended targets in the environment, each target is again sampled rhythmically, but the sampling frequency is slower, often falling in the lower end of the theta frequency band (~4 Hz) (*Fiebelkorn et al., 2013*; *Landau and Fries, 2012*; *Re et al., 2019*). When the attended targets appear in different visual hemifields, an alternating sampling strategy was observed, evidenced by the 180-degree phase relationship between the two behavioral time courses (*Chota et al., 2022*; *Fiebelkorn et al., 2018*).

How distractors are temporally sampled has not been investigated to date. One of the reasons is that distractors do not elicit behavioral responses, and as such, a pure behavioral approach is not able to address this question. We overcame the problem by recording EEG in an SSVEP paradigm in which the target and the distractor overlapped in space and time and flickered at different frequencies, a method referred to as frequency tagging. Separately extracting the EEG signals underlying the neural response to the target and that to the distractor according to their flickering frequencies (4.29 Hz for target and 6 Hz for distractor), we found that at the whole trial level, target processing exhibited higher SSVEP amplitude than distractor processing, and for both target and distractor processing, the signal power is maximal at the posterior channels. Cognizant of the possibility that the power at 4.29 Hz may leak into neighboring frequency bands where the power is weaker (see Appendix 1—figures 4), instead of using the 6 Hz SSVEP amplitude to quantify distractor processing, we adopted the MVPA decoding approach to quantify the distractor processing by leveraging the previous finding that different categories of emotional images evoked different patterns of neural responses in scalp EEG (*Bo et al., 2022*). This led us to construct classifiers that took the 6 Hz SSVEP amplitude across all electrodes as input to decode the spatial patterns evoked by different categories of emotional distractors, with higher classification or decoding accuracy taken to indicate stronger distractor processing. At the whole trial level, the observed above-chance decoding accuracy suggested that the distractor information is present in the brain and could be revealed and quantified by combining machine learning with distractor-specific scalp EEG.

Prior studies of visual environmental sampling used the cue-target paradigm in which the cue serves both to instruct the participant on how the target should be responded to and to reset the brain oscillation mediating the rhythmic visual sampling (*Kayser, 2009*). In our paradigm, the resetting was prompted by the onset of the compound stimulus array. The time elapsed after the stimulus array onset, referred to as TFO here, plays the role of the SOA in the cue-target paradigm. To index the temporal dynamics of target and distractor processing, we applied a moving window approach, in which the window duration was 0.5 s, and the step size was 0.25 s. Within each window, the 4.29 Hz SSVEP amplitude was taken to index target processing and the accuracy of decoding different categories of emotional distractors based on the 6 Hz SSVEP amplitude pattern was taken to index distractor processing. Plotting these two indices as functions of TFO, we assessed the temporal dynamics of target and distractor processing. The results revealed that both the target and the distractor were sampled rhythmically with the same sampling frequency of ~1 Hz (delta frequency band), which is considerably slower than those reported in previous studies (*Re et al., 2019*) in which the sampling frequency tends to fall in the theta frequency band (4–8 Hz).

Delta oscillations (0.5–3.5 Hz), traditionally associated with deep sleep and homeostatic processes (*Amzica and Steriade, 1998*; *Franken et al., 2001*; *Franken and Dijk, 2024*; *Frohlich et al., 2021*; *Torres-Herraez et al., 2022*), are being increasingly recognized for their role in a variety of cognitive functions (*Basar et al., 1999*; *Başar et al., 2001*; *Başar-Eroglu et al., 1992*). In the auditory domain, rhythmic sampling of an auditory scene is shown to be mediated by delta oscillations (*Kubetschek and Kayser, 2021*; *Morillon et al., 2019*). Our findings suggest that similar mechanisms could also

operate in the visual domain. In a recent study, when observers directed temporal attention to one of two sequential grating targets with predictable timing, the steady-state visual evoked response of the flashing target was modulated at 2 Hz (*Denison et al., 2022*), which falls in the delta frequency band. In addition, extensive evidence has shown that expecting a stimulus, which is known to require the deployment of attentional resources, engages delta oscillations (*Arnal et al., 2011*; *Arnal et al., 2015*; *Breska and Deouell, 2017*; *Cravo et al., 2013*; *Lakatos et al., 2008*; *Schroeder and Lakatos, 2009*; *Stefanics et al., 2010*). Delta oscillations were also involved in mechanisms that synchronize distributed regions within functional neural networks in supporting cognitive control (*Helfrich et al., 2017*; *Helfrich et al., 2019*; *Helfrich and Knight, 2016*). The spatially overlapping target and distractor in our paradigm places high demand on the brain's cognitive control system, shown recently to be operating in the delta frequency band (*Pagnotta et al., 2024*), which could be another reason underlying the observed mediation by delta oscillations in the rhythmic sampling of the target-distractor environment.

## Phase relationship between target and distractor sampling and its functional significance

As mentioned earlier, when two attended objects are presented simultaneously in different visual hemifields, the visual system tends to sample them in a serial, alternating fashion, as evidenced by two rhythmic behavioral time series exhibiting a 180-degree relative phase (antiphase) (*Denison et al., 2022*; *Fiebelkorn et al., 2013*; *Mo et al., 2019*). When two attended objects overlap in space, however, this alternating sampling pattern is not observed, and the relative phase between the two rhythmic behavioral time series appears to be uniformly distributed across participants (*Re et al., 2019*). In our experimental design, the target and the distractor overlapped in space, which is a configuration known to maximize the distraction effect, and the relative phase between the rhythmic samplings of the target and the distractor is also uniformly distributed across participants. Thus, regardless of the behavioral relevance of the two superimposed stimuli, there is no preferred phase relationship between their samplings at the population level.

   Although a clear phase relationship between the target sampling and the distractor sampling is absent at the population level, the relative phase between the two time series may nonetheless have functional significance. In particular, when the target sampling and the distractor sampling occur in opposite phases of a sampling cycle, i.e., when they are 180 degrees out of phase, the interference should be minimized, and consequently, the task performance should be maximized. On the contrary, when the target sampling and the distractor sampling occur in the same phase of a sampling cycle, i.e., when the target and the distractor are sampled at the same time, the interference should be maximized, and the task performance should be minimized. Our results supported this hypothesis. Specifically, we showed that there was a positive correlation between the relative phase of the target and distractor sampling time series and the behavioral performance, namely, the greater the relative phase between the two time series, the higher the rate of correctly detecting the instances of coherent motion in the moving dots (attended target). The additional significance of this finding can be understood by considering the analysis results at the whole trial level. One may expect that at the whole trial level, the stronger the distractor representation indexed by higher decoding accuracy, the worse the task performance. This turned out to be not the case. As shown in *Figure 3D*, the distractor decoding accuracy at the whole trial level was not correlated with task performance, nor was the overall power of the target evoked activity at the whole trial level. Thus, what we found should be considered a new mechanism underlying the competition between distractor and target. In this mechanism, the key is not how well the target and the distractor are each represented but how their respective rhythmic sampling aligns over time: The more target sampling and distractor sampling are separated in time, the less direct competition between the two, the better the attended information is processed, and the better the behavioral performance.

## Signal processing considerations

First, when the amplitude of a periodic signal with a frequency f is modulated at 1 Hz, we should observe sidebands at f+1 and f−1 Hz in the Fourier spectrum of the signal. These sidebands are not clearly seen in the Fourier spectrum of the SSVEP time series (see *Figure 2B*). We investigated the underlying reason in Appendix 1. The starting point is the observation that biological data is noisy.

The SSVEP from the subjects contains a varying amount of noise quantified by the signal-to-noise ratio (SNR). We showed using both simulations and actual data that when the SNR is high, the sidebands are visible, whereas when the SNR is low, the sidebands are indistinguishable from the noise floor (*Appendix 1—figure 1*). The majority of our subjects have low SNR for observing sidebands. This is why the sidebands in the Fourier spectrum in *Figure 2B* are not readily identifiable. Second, when a 4.29 Hz periodic component and a 6 Hz component are combined, one should observe a beating frequency at 1.71 Hz. This beating frequency is clearly seen in the Fourier spectra of the amplitude envelope of the SSVEP in *Figure 4A*. However, this spectral peak is secondary to a much stronger spectral peak occurring at ~1 Hz, which cannot be explained from a pure signal processing perspective (see *Appendix 1—figure 2* for further investigation). This suggests that the 1 Hz amplitude modulation of the SSVEP amplitude, as well as decoding accuracy time series, is of an endogenous origin and represents the frequency of the rhythmic sampling of the environment by the visual attention system in our paradigm. Third, we tested the effect of moving window parameters on the temporal dynamics of target and distractor processing. Using a 0.1 s window length and a 0.05 s step size (*Appendix 1—figures 5 and 6*) and applying the window-free Hilbert transform method (*Appendix 1—figures 7 and 8*), we found the same results as those reported in the main manuscript, suggesting that the ~1 Hz rhythmic sampling and the phase-related target-distractor competition are robust findings. Fourth, to further test the robustness of the decoding results, we implemented a random permutation procedure. *Appendix 1—figure 9* shows the results based on 1000 permutations. For each of the three pairwise classifications—pleasant vs neutral, unpleasant vs. neutral, and pleasant vs. unpleasant—as well as the three-way classification, the actual decoding accuracies fall far outside the null-hypothesis distribution (p<0.001), and the effect sizes are extremely large.

## Limitations

First, the experimental paradigm lacked a no-distractor baseline condition. The SSVEP amplitude of the target at the whole trial level thus reflected the combined effect of the stimulus parameters (e.g. contrast of the moving dots), as well as attention. However, the time course of the target SSVEP amplitude within a trial, derived from the moving window analysis, reflected the temporal fluctuations of target processing, since the stimulus parameters remained the same during the trial. Second, target processing and distractor processing are quantified differently: SSVEP amplitude for the former and decoding accuracy for the latter. However, using SSVEP amplitude to quantify target processing is a well-established approach, and given that decoding is between different classes of distractors, we are also confident that the decoding accuracy reflects distractor processing. For comparing the two, we normalized each time course to make them dimensionless and then computed correlations. Third, no fusion was attempted between simultaneously recorded EEG and fMRI. However, given that this study concerns the temporal dynamics of target and distractor processing, it is felt that fMRI data, which is known to possess low temporal resolution, has limited potential to contribute.

## Summary

In this work, we reported two main findings: (1) in sustained visual attention under distraction, both the distractor and the target are sampled rhythmically, with the sampling frequency being ~1 Hz (i.e. in the delta frequency band) and (2) the temporal relationship between the distractor sampling and the target sampling is a significant factor underlying task performance with a more antiphase relationship giving rise to better behavioral performance. To further illustrate the importance of the second finding, we note that neither target nor distractor processing strength at the whole trial level correlates with behavioral accuracy. These results extend the rhythmic sampling theory to distractor processing and provide further support for the important role of low-frequency brain oscillations in organizing cognitive operations. They also demonstrate the utility of applying machine learning methods in uncovering the temporal dynamics of sustained attention in target-distractor scenarios.

# Materials and methods
## Participants

The experimental protocol was approved by the Institutional Review Board of the University of Florida. Thirty undergraduate students from the University of Florida gave written informed consent

and participated in the experiment to earn credit in an introductory psychology course. Because the EEG data were recorded inside the MRI scanner (simultaneous EEG-fMRI), participants underwent screening for ferromagnetic implants, claustrophobia, and personal or family history of epilepsy or photic seizures. Female participants were also administered a pregnancy test before participation. Three participants were excluded due to excessive movements during recording. The EEG data from n=27 participants (18 women, 9 men, mean age = 19.2 ± 1.1 years) were analyzed and reported here.

## Stimuli

The stimulus comprised an RDK overlaid on affective images selected from the IAPS database. The RDK consisted of 175 yellow dots randomly distributed within a circular aperture in the center of the screen, with each dot spanning <0.5 degrees of visual angle. The IAPS images portrayed three broad categories of emotions: pleasant, neutral, and unpleasant. They were similar in overall composition and rated complexity and matched in picture file size to minimize confounds across categories. The stimulus was presented on a 30-inch MR-compatible LCD monitor placed approximately 230 cm from the participant's head outside the bore of the MRI scanner. A white fixation dot was displayed at the center of the screen throughout the experiment.

## Procedure

See *Figure 1A* for the schematic illustration of the experimental task. After 5–11 s of fixation, the participant was presented the compound stimulus array consisting of the randomly moving dots (RDK) superimposed on the IAPS pictures for a duration of 11.667 s. The moving dots and the background pictures were flickered on and off at 4.29 Hz and 6 Hz, respectively. For each 4.29 Hz flicker cycle, the moving dots were displayed for 100 ms, which was followed by a 133 ms off period. Similarly, for each 6 Hz flicker cycle, the IAPS background picture was shown for 100 ms and followed by a 66.7 ms off period. During each on-off cycle, the moving dots in the RDK were randomly displaced by 0.3 degrees of visual angle in either random directions or one coherent direction. Coherent motion instances lasted for four on-off cycles (933 ms) and appeared once in 39 trials (13 trials per emotion condition) or twice in 4 trials. The remaining 41 trials contained no instances of coherent motion. Each trial lasted 11.667 s (50 moving dot cycles and 70 IAPS background picture cycles). The coherent motion instances occurred in the interval between 2.3 s and 10.4 s post stimulus array onset. The participant was asked to fixate on the central white dot during the trial, to monitor the motion coherence of the random dots, and report the number of coherent motion instances at the end of the trial. Both the number of coherent motion instances and the underlying emotion category of IAPS image were randomized in each trial. A total of 42 IAPS pictures were equally divided into three content categories based on valence: pleasant (erotic couples), neutral (workplace people), or unpleasant (bodily mutilation). Depending on the emotion category of the IAPS picture used in a given trial, the trials are referred to as pleasant, neutral, and unpleasant trials. There was a total of 84 trials: 28 pleasant trials, 28 unpleasant trials, and 28 neutral trials, and each picture was used twice during the experiment.

## EEG data collection and preprocessing

EEG data was recorded using a 32-channel MR-compatible EEG recording system (Brain Products, Germany). The system was synchronized to the internal clock of the scanner to facilitate the subsequent scanner noise removal. Thirty-one Ag/AgCl electrodes were placed on the scalp according to the 10–20 system via an elastic cap. One additional electrode was located on the participant's upper back to record the electrocardiogram (ECG). Electrode FCz was used as the reference during recording. Impedances were kept below 20 kΩ for all scalp electrodes and below 50 kΩ for the ECG electrode, as suggested by the manufacturer. EEG data was digitized at 16-bit resolution and sampled at 5 kHz with a 0.1–250 Hz (3 dB point) bandpass filter applied online (Butterworth, 18 dB/octave roll off). The digitized data was transferred to a laptop computer via a fiber-optic cable.

Artifact removal from EEG data, specifically the removal of magnetic gradient and cardioballistic artifacts, was conducted using the Brain Vision Analyzer 2.0 software (Brain Products GmbH). The elimination of magnetic gradient artifacts was based on an algorithm initially proposed by *Allen et al., 2000*. The process involves the creation of an artifact template through averaging EEG data over 41 consecutive fMRI volumes, which was subsequently subtracted from the EEG recordings. Additionally, cardioballistic artifacts were removed by employing a technique developed by *Allen et al., 1998*, in

which R peaks were detected via the EKG electrode, and a corrective template was computed from 21 successive heart beats and subtracted from the EEG data.

Subsequent to scanner artifact removal, data was downsampled to 500 Hz and exported into EEGLab software (*Delorme and Makeig, 2004*). The data underwent further filtering using a 0.1–40 Hz band-pass Butterworth filter. Independent components analysis was applied to remove components associated with eye blinks, horizontal eye movements, and residual cardioballistic artifacts. The data were then converted to the average reference.

## EEG data analysis

### Overview

According to the task design, the IAPS pictures were behaviorally irrelevant and thus the distractor to be ignored, while the moving dots were behaviorally relevant and thus the target to be attended. To minimize the transient effect resulting from the stimulus array onset and the possible effect resulting from anticipating the end of a trial, the EEG data from the beginning and the end of a trial were discarded, namely, the analyzed EEG data came from the period from 2 to 11 s post array onset, which contained the period from 2.3 to 10.4 s post stimulus array onset during which instances of coherent motion in the moving dots took place.

### Quantifying target processing

The moving dots were flickered at 4.29 Hz. For a given type of emotion trials (i.e. pleasant, neutral, or unpleasant), the SSVEP was computed by averaging all the trials within the type. Filtering the SSVEP between 4.29–0.5 Hz and 4.29+0.5 Hz yielded the data specific for target processing. Obtaining the magnitude of the band-pass filtered data at the whole trial level allowed the assessment of the overall strength of target processing; see *Figure 1B*. To assess target processing as a function of TFO, i.e., the temporal dynamics of target processing, the magnitude of the band-passed filtered data was obtained using a moving window approach, where the window duration was 0.5 s and the step size was 0.25 s. See *Figure 1B* for illustration.

### Quantifying distractor processing

The IAPS pictures were flickered at 6 Hz. Band-pass filtering the EEG data between 6–0.5 Hz and 6+0.5 Hz resulted in signals that were specific to distractor processing. Following a recent study where we showed that the emotion category of IAPS pictures can be decoded from scalp EEG data using the MVPA method (*Bo et al., 2022*), we assessed distractor processing using an MVPA decoding approach at the whole trial level, as well as at the level of moving windows. The MVPA analysis was conducted with the linear support vector machine (SVM) method as implemented in the LibSVM package (http://www.csie.ntu.edu.tw/~cjlin/libsvm/) (*Chang and Lin, 2011*). The decoding was between two types of emotion trials (e.g. pleasant vs neutral) or between all three types of emotion trials based on a one-vs-all strategy. Above-chance decoding accuracy (50% for pairwise decoding and 33.3% for three-way decoding) is taken to indicate distractor processing in the brain with higher decoding accuracy indicating stronger distractor processing. For both the whole trial and moving window analysis, the trials from each of the three different emotional categories were divided into three subsets of trials randomly. We averaged the trials within each subset to yield the subset SSVEP. For the whole trial analysis, we calculated the 6 Hz SSVEP over the whole trial, whereas for the moving window analysis, the 6 Hz SSVEP amplitude was obtained for each 0.5 s analysis window. For the decoding strategy, the SSVEP amplitude from the two subsets within each emotion category served as training data for constructing an SVM classifier, while the SSVEP amplitude from the third subset was used as testing data for calculating decoding accuracy. This process was iterated 100 times to ensure the stability of the decoding result, and the average of the decoding accuracy values was analyzed and reported (*Bae and Luck, 2019*; *Haxby et al., 2014*; *Zhang et al., 2024*). See *Figure 1B* for an illustration of the method.

### Quantifying the relationship between target and distractor processing

To investigate the temporal relationship between target processing and distractor processing, we calculated the phase relationship between the target amplitude time series from the moving window

approach which quantified the temporal fluctuation of the strength of target processing and the distractor decoding accuracy time series which quantified the temporal fluctuation of the strength of distractor processing. To investigate the effect of temporal competition between target processing and distractor processing, we correlated the relative phase relationship between the target processing time series and the distractor processing time series with behavioral performance.

## Acknowledgements

This work was supported by NSF grants BCS2318886 and BCS2318984 and NIH grant R01 MH125615.

## Additional information

### Funding

| Funder | Grant reference number | Author |
|---|---|---|
| U.S. National Science Foundation | BCS2318886 | Andreas Keil Mingzhou Ding |
| U.S. National Science Foundation | BCS2318984 | Andreas Keil Mingzhou Ding |
| National Institutes of Health | R01 MH125615 | Andreas Keil Mingzhou Ding |

The funders had no role in study design, data collection and interpretation, or the decision to submit the work for publication.

### Author contributions

Changhao Xiong, Conceptualization, Software, Formal analysis, Validation, Investigation, Visualization, Methodology, Writing - original draft, Writing – review and editing; Nathan M Petro, Data curation; Ke Bo, Resources, Data curation, Investigation, Visualization, Methodology; Lihan Cui, Investigation, Visualization, Methodology; Andreas Keil, Resources, Data curation, Supervision, Writing – review and editing; Mingzhou Ding, Conceptualization, Supervision, Investigation, Project administration, Writing – review and editing

### Author ORCIDs

Changhao Xiong http://orcid.org/0009-0008-2766-8373
Nathan M Petro https://orcid.org/0000-0002-7553-4459
Ke Bo https://orcid.org/0000-0003-3286-7891
Lihan Cui https://orcid.org/0000-0001-7722-3550
Andreas Keil https://orcid.org/0000-0002-4064-1924
Mingzhou Ding https://orcid.org/0000-0002-1024-3503

### Ethics

Human subjects: The experimental protocol was approved by the Institutional Review Board of the University of Florida. All participants gave written informed consent before participating in the study.

Reviewer #1 (Public review): https://doi.org/10.7554/eLife.106140.3.sa1
Reviewer #2 (Public review): https://doi.org/10.7554/eLife.106140.3.sa2
Author response https://doi.org/10.7554/eLife.106140.3.sa3

## Additional files

### Supplementary files

MDAR checklist

## Data availability

The data used in this study have been uploaded at Dryad and can be accessed via the link: https://doi.org/10.5061/dryad.xd2547dw5.

The following dataset was generated:

| Author(s) | Year | Dataset title | Dataset URL | Database and Identifier |
|---|---|---|---|---|
| Xiong C, Petro N | 2025 | RDK-IAPS paradigm EEG, target vs distractor | https://doi.org/10.5061/dryad.xd2547dw5 | Dryad Digital Repository, 10.5061/dryad.xd2547dw5 |

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

# Appendix 1

In this appendix, we considered several issues related to this study: (1) signal processing issues underlying the spectral analysis of various time series data, (2) quantifying distractor processing with MVPA decoding, (3) effect of moving window parameters, and (4) robustness of decoding analysis.

## Signal processing issues

The power of the 4.29 Hz (the target) and 6 Hz (the distractor) signals are both modulated at around 1 Hz. In the Fourier spectrum, the sidebands should be visible at around 3.29 Hz and 5.29 Hz (4.29 Hz±1 Hz), as well as at around 5 Hz and 7 Hz (6 Hz±1 Hz). However, there are no clear peaks visible at these frequencies from *Figure 2B*. We examine the reason here.

For clean sinusoidal signals with periodic amplitude modulation, we should observe sidebands. However, biological data is noisy, and the SSVEP from each subject shows significant variability in SNR (see definition below). SNR determines whether we can observe sideband frequencies or not. We demonstrate this point first through simulation and then on our data.

## Simulation

Simulated signals were generated by adding a 4.29 Hz component and a 6 Hz component together. These were the same frequency components as in our experiment and also, similar to what we observed in the data, the magnitude of the 6 Hz component was made ½ that of the 4.29 Hz component. We then modulated the amplitude of these components at 1 Hz (the same as that observed in our data). The time course is shown in the left panel of *Appendix 1—figure 1*. In the right panel of *Appendix 1—figure 1*, we showed the Fourier spectrum, where the sidebands for both the 4.29 Hz component (at 3.29 Hz and 5.29 Hz) and the 6 Hz component (at 5 Hz and 7 Hz) are clearly seen. Note that there was no noise in this case.

Next, we added noise to the same signal. The SNR is defined as:

$$SNR_{dB} = 10 \cdot log_{10} \frac{P_{signal}}{P_{noise}}$$

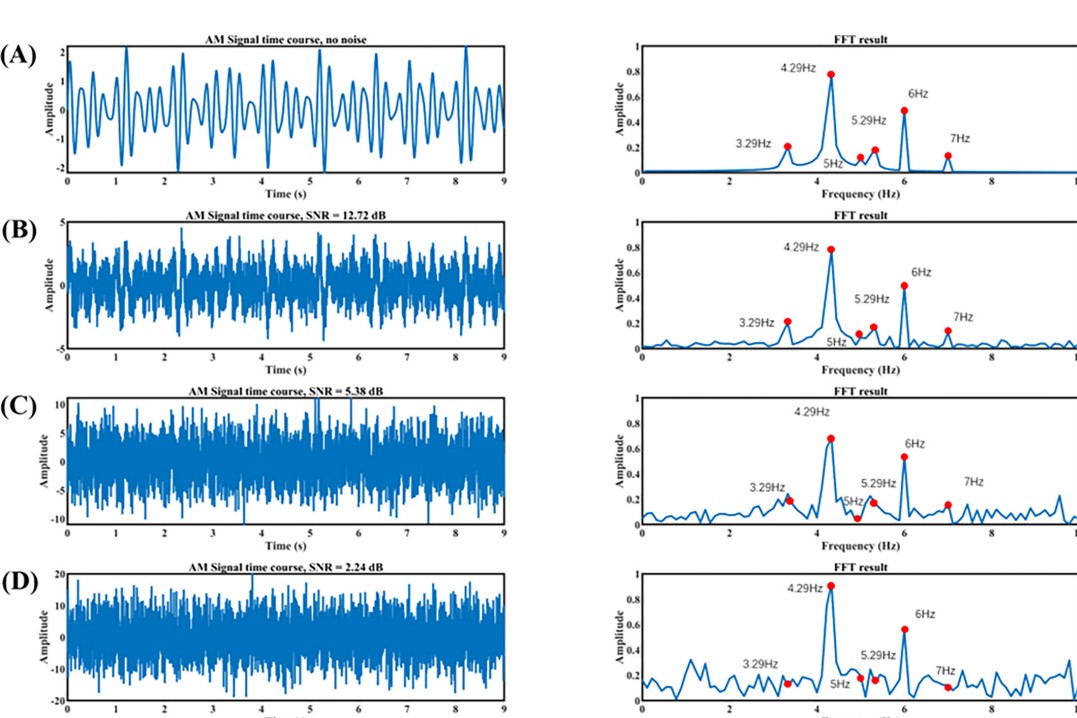

**Appendix 1—figure 1.** Simulation results. (**A**) The signal containing a 4.29 Hz component and a 6 Hz component where the 6 Hz signal's magnitude is about half that of the 4.29 Hz signal. The amplitude is modulated at 1 Hz. No noise is added. (**B**) Low level of noise is added to the signal in *Appendix 1—figure 1A*, where the signal-to-noise ratio (SNR) = 12.72 dB. Sidebands are still seen. (**C**) Middle level of noise is added to the signal in *Appendix 1—figure 1A* where the SNR = 5.38 dB. Sidebands become difficult to see. (**D**) High level of noise is added to the

signal in **Appendix 1—figure 1A** where the SNR = 2.24 dB, sidebands become more indistinguishable from the noise floor. Red dots indicate the location of the main frequency components and the locations where the sidebands should appear.

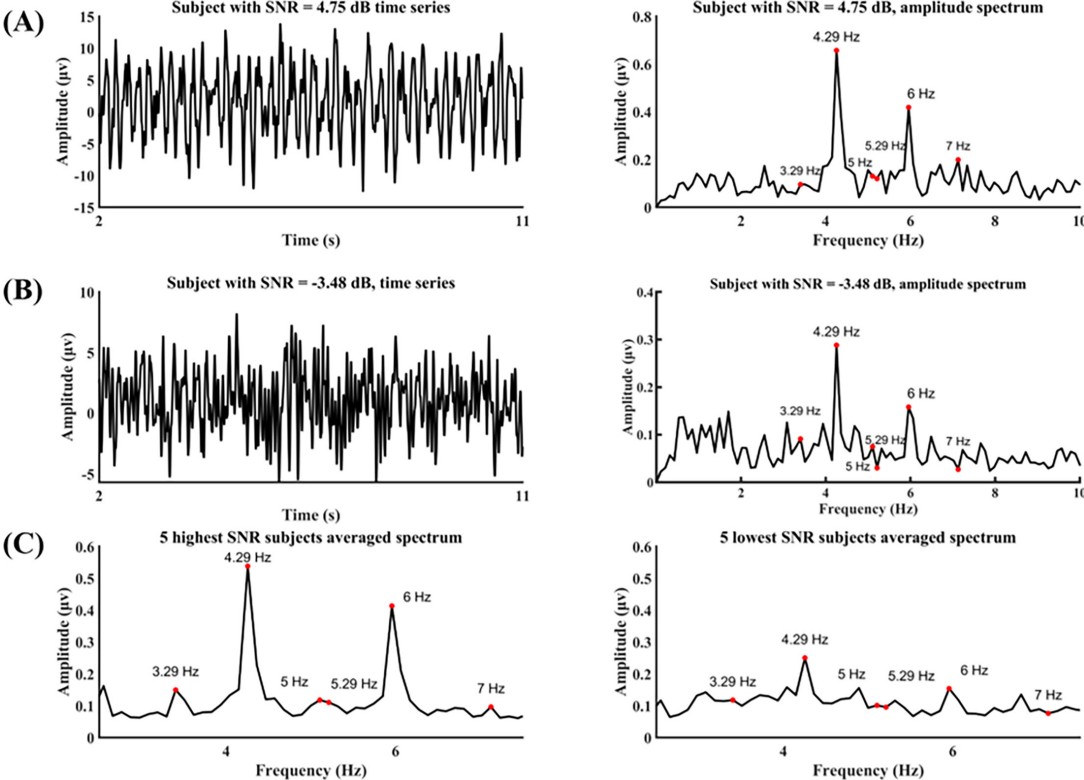

**Appendix 1—figure 2.** Experimental data. (**A**) The time course of the steady-state visual evoked potential (SSVEP) and its Fourier spectrum from a subject with high signal-to-noise ratio (SNR). The sidebands can be observed. (**B**) The time course and its Fourier spectrum from a subject with low SNR. The sidebands are indistinguishable from the noise floor. (**C**) The averaged Fourier spectrum from five highest SNR subjects and five lowest SNR subjects. Again, for subjects with high SNR, the sidebands are identifiable, whereas for subjects with low SNR, the sidebands are not identifiable.

where the signal power is defined to be the average Fourier power within 3.8–4.8 Hz and 5.5–6.5 Hz and the noise power is the average Fourier power within 0–3 Hz and 7–10 Hz. **Appendix 1—figure 1B**, **Appendix 1—figure 1C**, and **Appendix 1—figure 1D** show the results after adding progressively more noise to the simulated signal. When the noise level is low, e.g., SNR = 12.72 dB (**Appendix 1—figure 1B**), the sidebands are still clearly visible, as shown in the right panel of **Appendix 1—figure 1B**, although they are not as prominent as in the right panel of **Appendix 1—figure 1A**. When more noise is added, as shown in **Appendix 1—figure 1C**, where SNR = 5.38 dB, which is similar to what we see in a typical high SNR subject in our data, the sidebands are beginning to become indistinguishable from the noise floor. **Appendix 1—figure 1D** shows a case when an even higher level of noise is added, e.g., SNR = 2.24 dB, the sidebands become even more indistinguishable from the noise floor.

## Experimental data

Now we demonstrate the impact of SNR on sidebands in experimental data. **Appendix 1—figure 2A and B** compares two subjects, one with relatively high and the other relatively low SNRs, respectively. For the subject with high SNR, the sidebands are still somewhat distinguishable from the noise floor, whereas for the subject with lower SNR, the sidebands are no longer visible. In **Appendix 1—figure 2C**, we averaged the Fourier spectra of the five subjects with the highest SNR and that of the five subjects with the lowest SNR, and the results again indicate that SNR plays a major role in determining whether the sidebands can be seen or not. For the Fourier spectrum averaged across all

subjects, which is the figure shown in the manuscript, because of the influence of low SNR subjects, the sidebands are not clearly visible.

## Decoding distractors

In the SSVEP literature, signal amplitude at the flicker frequency is the main variable for quantifying the processing of the flickering stimulus. In this work, we treated the scalp pattern of the signal amplitude at the distractor flicker frequency as input features and subjected them to decoding analysis. Below we show that the decoding accuracy is a more suitable variable for quantifying distractor processing.

### Whole trial analysis

The target amplitude (4.29 Hz) and the distractor amplitude (6 Hz) were extracted from each subject and displayed in *Appendix 1—figure 3A*. The strong correlation suggests that there is significant power leakage from the stronger target frequency into the weaker distractor frequency. However, when the target amplitude and distractor decoding accuracy are plotted, no correlation was found, as shown in *Appendix 1—figure 3B*, suggesting that the target amplitude has no influence on distractor decoding accuracy.

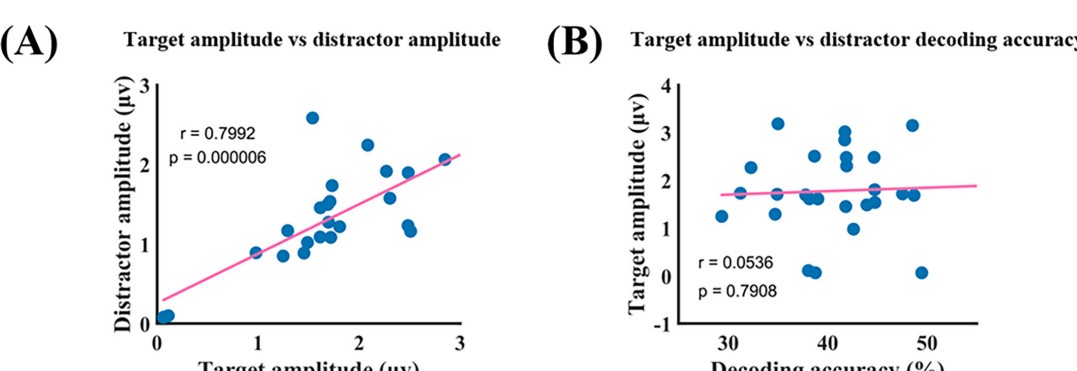

**Appendix 1—figure 3.** Steady-state visual evoked potential (SSVEP) amplitude analysis at the whole trial level. (**A**) Target amplitude vs distractor amplitude, where the correlation is r=0.7992 (p=0.000006), suggesting the 6 Hz signal amplitude is strongly influenced by the 4.29 Hz signal amplitude. (**B**) Target amplitude vs distractor decoding accuracy, where the correlation is r=0.0536 (p=0.7908), suggesting that the decoding accuracy as an index of distractor processing is not influenced by the 4.29 Hz target amplitude.

### Moving window analysis

Similar results can be observed at the moving window level. The target amplitude time series and the distractor amplitude time series were extracted from each subject and the relationship between the two time series assessed. The relative phase is also correlated with behavior. *Appendix 1—figure 4A* shows that the relative phase is narrowly distributed around 0.23±0.05π, which is confirmed by the K-S statistic of 0.46 in *Appendix 1—figure 4B*, demonstrating that the relative phase distribution is significantly different from the uniform distribution at p=0.00001. This means that the power leakage from the target time series impacts the distractor time series, making it nearly phase locked to the target time series. It is not surprising that the relative phase between the two amplitude time series does not predict task performance (*Appendix 1—figure 4C*).

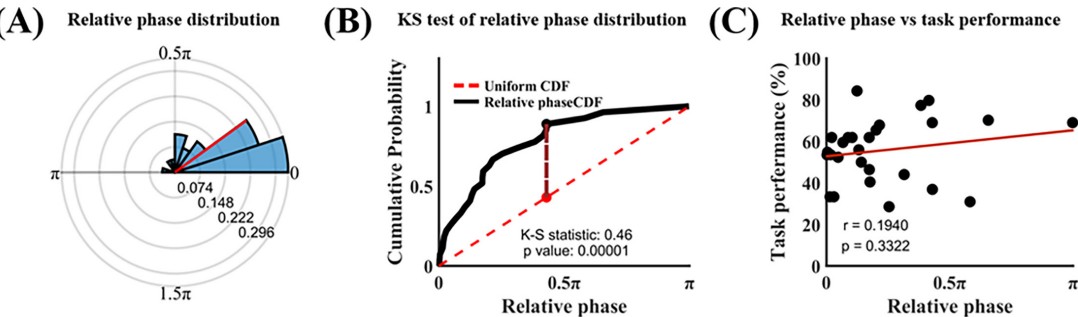

**Appendix 1—figure 4.** Moving window analysis. (**A**) The relative phase between the target amplitude time series and the distractor amplitude time series. (**B**) Kolmogorov-Smirnov test showed that the relative phase distribution is significantly different from the uniform distribution. (**C**) Relative phase vs task performance. r=0.1940 (p=0.3322) means that there is no significant correlation between amplitude relative phase and task performance.

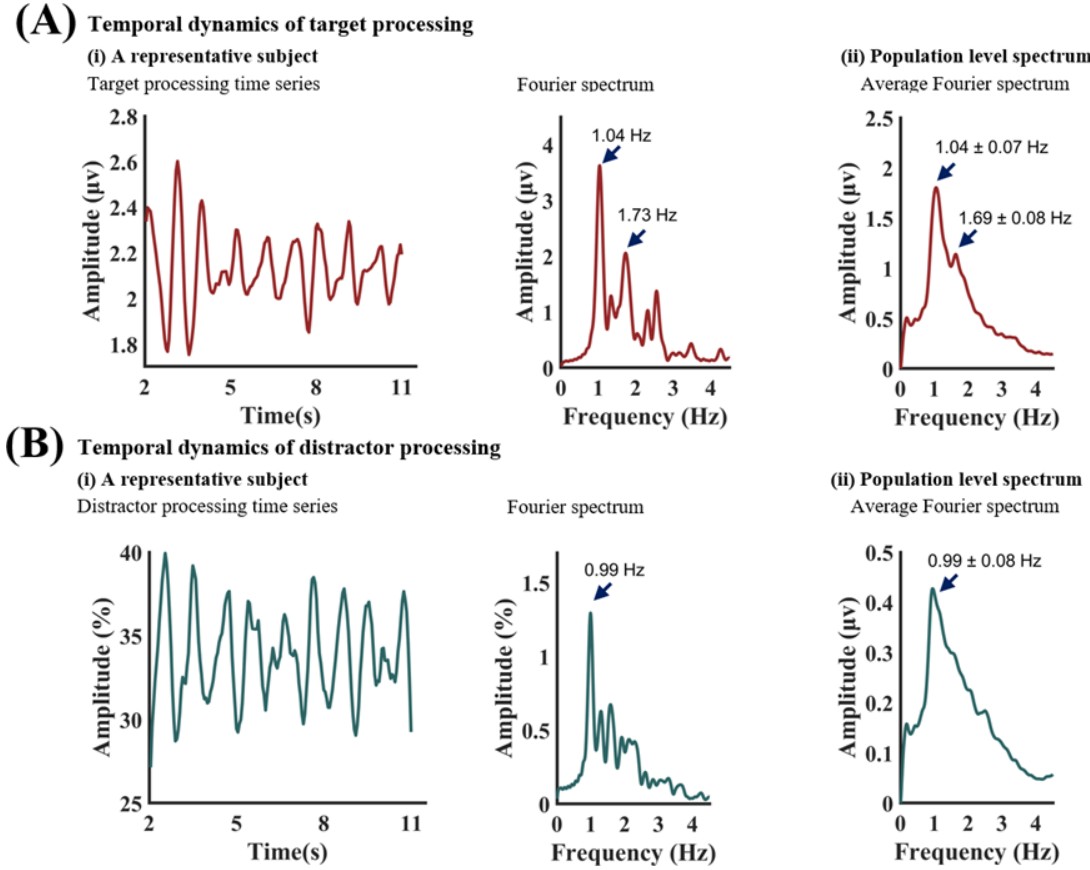

**Appendix 1—figure 5.** Temporal dynamics of target and distractor processing with 0.1 s window length and 0.05 s step size. (**A**) (**i**) Target processing time series from the moving window approach for a representative subject (left) and its Fourier spectrum (right). (**A**) (**ii**) The average Fourier spectrum across 27 subjects. (**B**) (**i**) Distractor processing time series from the moving window approach for a representative subject (left) and its Fourier spectrum (right). (**B**) (**ii**) The average Fourier spectrum across 27 subjects.

## Effect of moving window parameters

We redid the moving window analysis using a different set of windowing parameters, e.g., a 0.1 s sliding window length with a 0.05 s step size. *Appendix 1—figure 5* demonstrates that the strength of both target and distractor processing fluctuates around ~1 Hz, both at the individual and group levels. Additionally, *Appendix 1—figure 6* shows that the relative phase between target and distractor processing time series exhibits a uniform distribution across subjects. For the relationship

between relative phase and behavior, *Appendix 1—figure 6* illustrates two representative cases: a high-performing subject with 84.34% task accuracy exhibited a relative phase of 0.9483π, while a low-performing subject with 30.95% accuracy showed a phase of 0.29π. At the group level, a significant positive correlation between relative phase and task performance was found (r=0.6343, p=0.0004), as shown in *Appendix 1—figure 6*. All these results, aligning closely with our original findings, suggest that the conclusions are not dependent on windowing parameters.

To further validate our findings, we also employed the Hilbert transform to extract amplitude envelopes of the target and distractor signals on a time-point-by-time-point basis, providing a window-free estimate of signal strength (*Appendix 1—figure 7* and *Appendix 1—figure 8*). The results remain consistent with both the original findings and the new sliding window analyses (above). *Appendix 1—figure 7* reveals ~1 Hz fluctuations in target and distractor processing at both individual and group levels. *Appendix 1—figure 8* confirms a uniform distribution of the relative phase. As shown in *Appendix 1—figure 8*, the relative phase was 0.9567π for a high-performing subject (84.34% accuracy) and 0.2247π for a low-performing subject (28.57% accuracy). At the group level, a significant positive correlation was again observed between relative phase and task performance (r=0.4020, p=0.0376), as shown in *Appendix 1—figure 8*.

## Robustness of decoding analysis

To test robustness of the decoding analysis, we implemented a random permutation procedure in which trial labels were randomly shuffled to construct a null-hypothesis distribution of decoding accuracy. We then compared the decoding accuracy from the actual data to this distribution. *Appendix 1—figure 9* shows the results based on 1000 permutations. For each of the three pairwise classifications—pleasant vs neutral, unpleasant vs neutral, and pleasant vs unpleasant—as well as the three-way classification, the actual decoding accuracies fall far outside the null-hypothesis distribution (p<0.001), and the effect sizes are extremely large. These findings indicate that the observed decoding accuracies are statistically significant and robust in terms of both statistical inference and effect size.

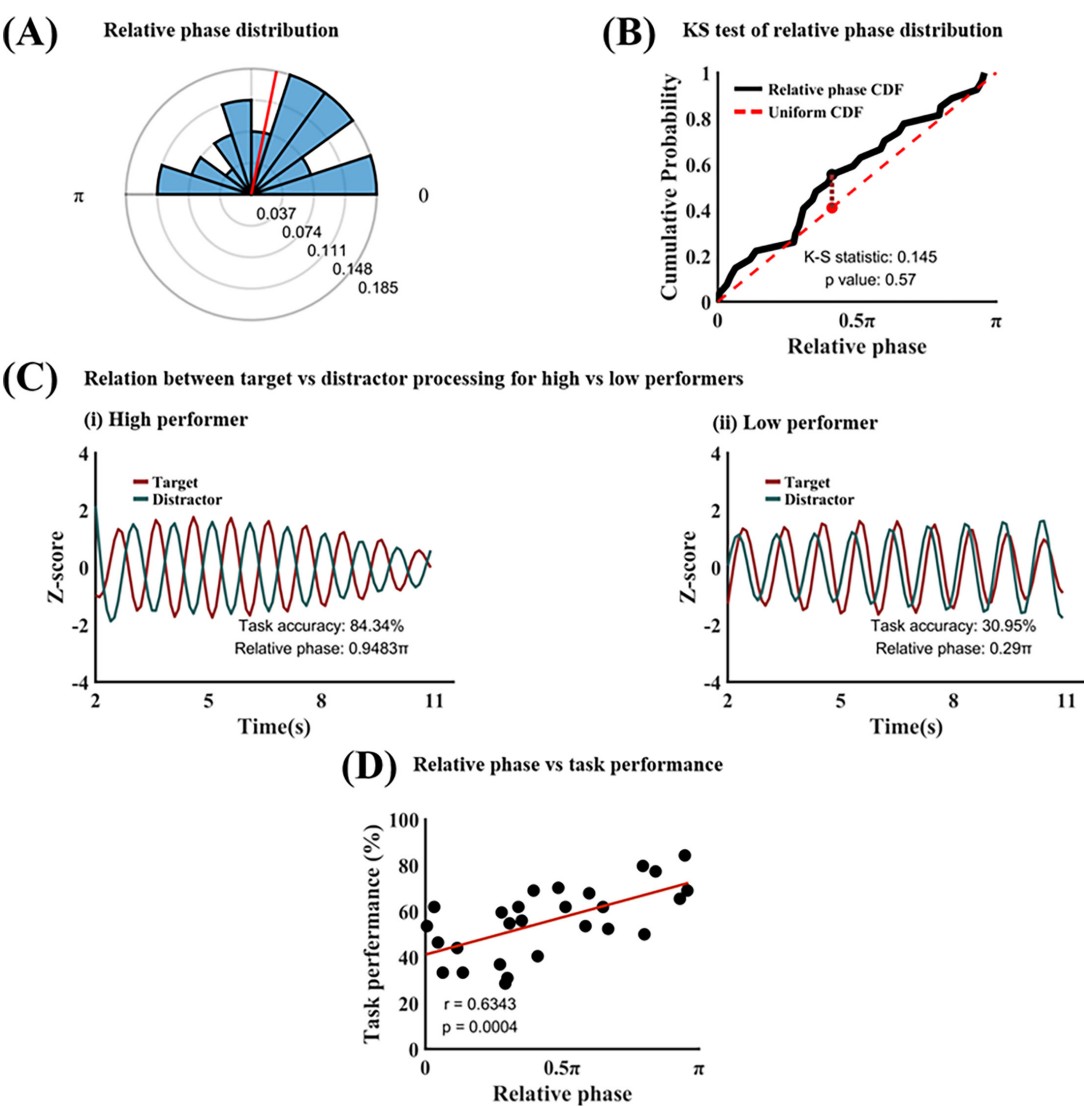

**Appendix 1—figure 6.** Target-distractor competition analysis with 0.1 s window length and 0.05 s step size. (A) Phase polar histogram for the relative phase between target process time series and distractor processing time series (1 Hz). The average relative phase is 0.44π. (B) Kolmogorov-Smirnov test showed that the relative phase distribution is not different from uniform distribution. (C) Temporal relationship between target processing and distractor processing for (i) a high performer (accuracy=83.84%; relative phase=0.9483π) and (ii) a low performer (accuracy=30.95%; relative phase=0.29π). (D) Task performance vs 1 Hz relative phase. The significant positive correlation (r=0.6343, p=0.0004) means that the more separated the target and distractor sampling within the 1 Hz oscillation cycle, the better the behavioral performance. CDF: cumulative distribution function

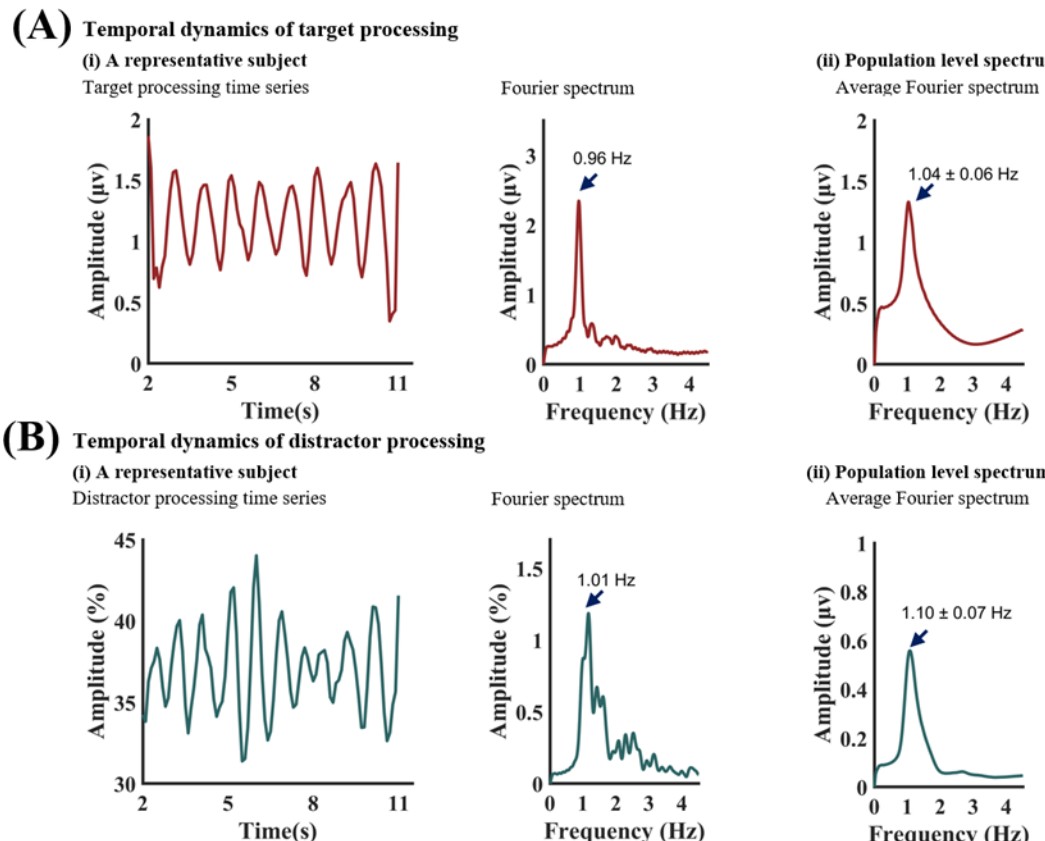

**Appendix 1—figure 7.** Temporal dynamics of target and distractor processing with Hilbert transformed target and distractor processing time series. (**A**) (**i**) Target processing time series from a representative subject (left) and its Fourier spectrum (right). (**A**) (**ii**) The average spectrum across 27 subjects. (**B**) (**i**) Distractor processing time series for a representative subject (left) and its Fourier spectrum (right). (**B**) (**ii**) The average spectrum across 27 subjects.

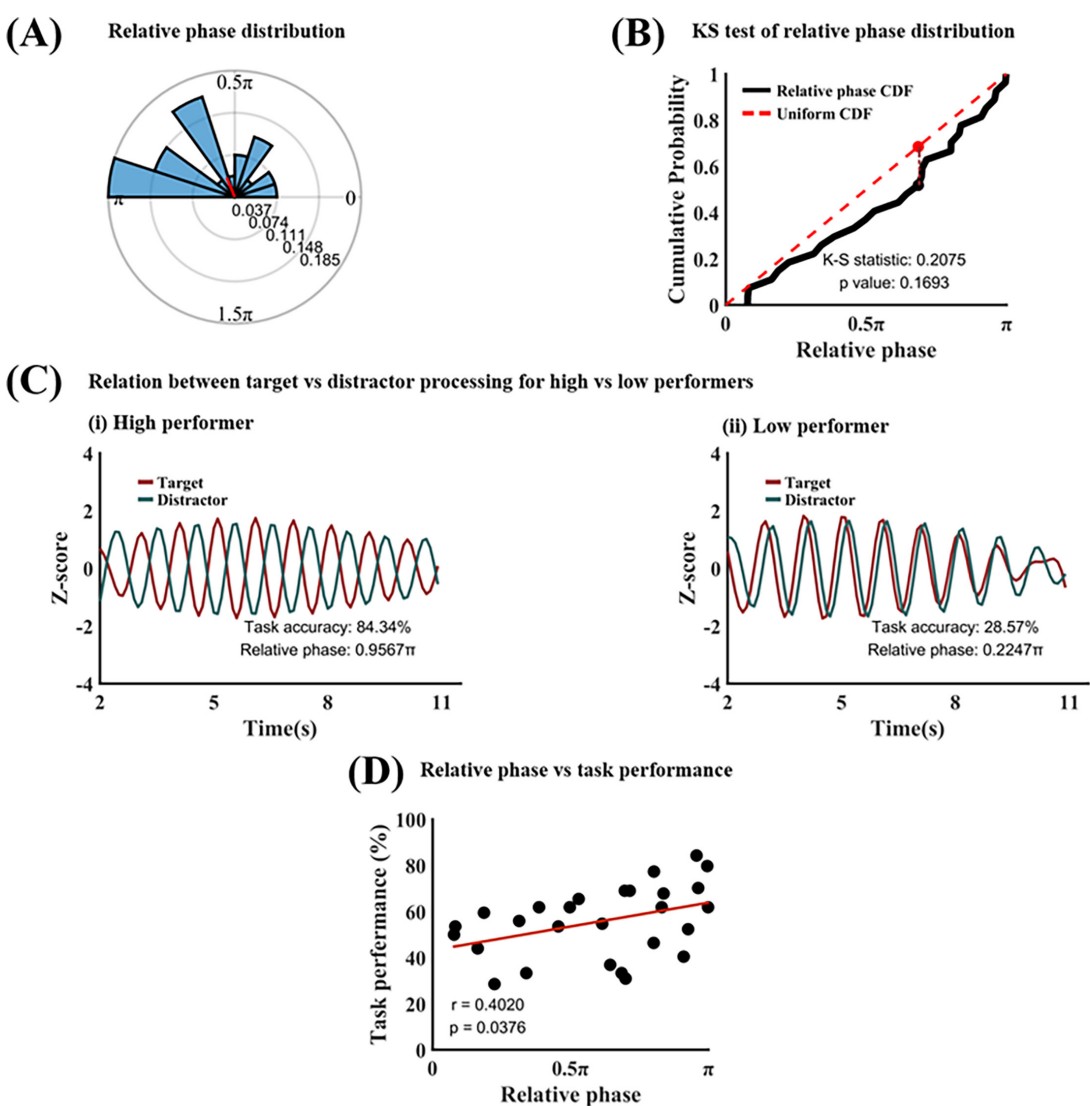

**Appendix 1—figure 8.** Target-distractor competition analysis with Hilbert transformed target and distractor processing time series. (**A**) Phase polar histogram for the relative phase between target process time series and distractor processing time series (1 Hz). The average relative phase is 0.63π. (**B**) Kolmogorov-Smirnov test showed that the relative phase distribution is not different from uniform distribution. (**C**) Temporal relationship between target processing and distractor processing for (**i**) a high performer (accuracy=83.84%; relative phase=0.9567π) and (**ii**) a low performer (accuracy=28.57%; relative phase=0.2247π). (**D**) Task performance vs 1 Hz relative phase. The significant positive correlation (r=0.4020, p=0.0376) means that the more separated the target and distractor sampling within the 1 Hz oscillation cycle, the better the behavioral performance. CDF: cumulative distribution function.

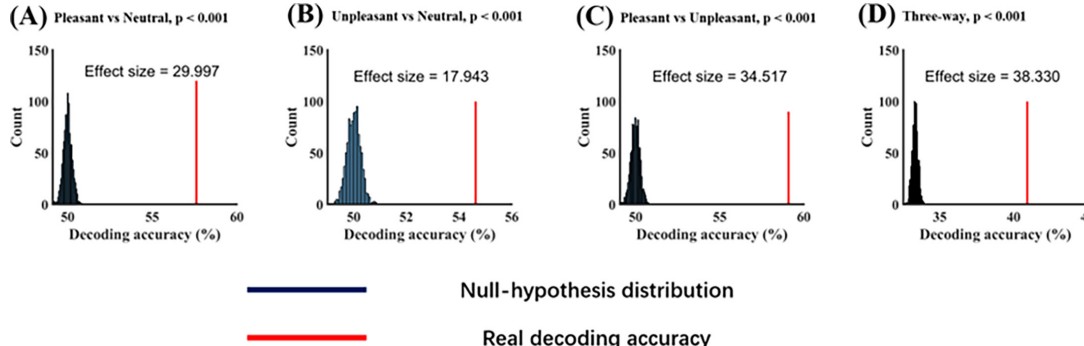

**Appendix 1—figure 9.** Comparison of actual decoding accuracy against the distribution of random permutation decoding accuracy. Random permutation decoding accuracy from (**A**) pleasant vs neutral, (**B**) unpleasant vs neutral, (**C**) pleasant vs unpleasant, and (**D**) three-way. In all four conditions, the actual decoding accuracy is significantly above chance level at p<0.001.

