## [Editor Report · eLife Assessment]

This work presents **important** information on rhythmicity of overlapping target and distractor processing and how this affects behaviour. The methods are, in general, clearly laid out and defensible, with several supplementary analyses leading to a **solid** base of evidence for their claims.

---

## [Referee Report · Reviewer #1 (Public review)]

Summary:

Using a combination of EEG and behavioural measurements, the authors investigate the degree to which processing of spatially-overlapping targets (coherent motion) and distractors (affective images) are sampled rhythmically and how this affects behaviour. They found that both target processing (via measurement of amplitude modulations of SSVEP amplitude to target frequency) and distractor processing (via MVPA decoding accuracy of bandpassed EEG relative to distractor SSVEP frequency) displayed a pronounced rhythm at ~1Hz, time-locked to stimulus onset. Furthermore, the relative phase of this target/distractor sampling predicted accuracy of coherent motion detection across participants.

Strengths:

- The authors are addressing a very interesting question with respect to sampling of targets and distractors, using neurophysiological measurements to their advantage in order to parse out target and distractor processing.

- The general EEG analysis pipeline is sensible and well-described.

- The main result of rhythmic sampling of targets and distractors is striking and very clear even on a participant-level.

- The authors have gone to quite a lot of effort to ensure the validity of their analyses, especially in the Supplementary Material.

- It is incredibly striking how the phase of both target and distractor processing are so aligned across trials for a given participant. I would have thought that any endogenous fluctuation in attention or stimulus processing like that would not be so phase aligned. I know there is literature on phase resetting in this context, the results seem very strong here and it is worth noting. The authors have performed many analyses to rule out signal processing artifacts, e.g. the sideband and beating frequency analyses.

Weaknesses:

- In general, the representation of target and distractor processing is a bit of a reach. Target processing is represented by SSVEP amplitude, which is going to most likely be related to the contrast of the dots, as opposed to representing coherent motion energy which is the actual target. These may well be linked (e.g. greater attention to the coherent motion task might increase SSVEP amplitude) but I would call it a limitation of the interpretation. Decoding accuracy of emotional content makes sense as a measure of distractor processing, and the supplementary analysis comparing target SSVEP amplitude to distractor decoding accuracy is duly noted. Overall, this limitation remains and has been noted in the Limitations section.

- Then comparing SSVEP amplitude to emotional category decoding accuracy feels a bit like comparing apples with oranges. They have different units and scales and reflect probably different neural processes. Is the result the authors find not a little surprising in this context? This relationship does predict performance and is thus intriguing, but I think this methodological aspect needs to be discussed further. For example, is the phase relationship with behaviour a result of a complex interaction between different levels of processing (fundamental contrast vs higher order emotional processing)? Again, this has been noted in the Limitations section, but changing the data to z-scores doesn't really take care of the conceptual issue, i.e. that on-screen contrast changes would necessarily be distracting during emotional category decision-making.

---

## [Referee Report · Reviewer #2 (Public review)]

In this study, Xiong et al. investigate whether rhythmic sampling - a process typically observed in the attended processing of visual stimuli - extends to task-irrelevant distractors. By using EEG with frequency tagging and multivariate pattern analysis (MVPA), they aimed to characterize the temporal dynamics of both target and distractor processing and examine whether these processes oscillate in time. The central hypothesis is that target and distractor processing occur rhythmically, and the phase relationship between these rhythms correlates with behavioral performance.

Major Strengths

(1) The extension of rhythmic attentional sampling to include distractors is a novel and interesting question.

(2) The decoding of emotional distractor content using MVPA from SSVEP signals is an elegant solution to the problem of assessing distractor engagement in the absence of direct behavioral measures.

(3) The finding that relative phase (between 1 Hz target and distractor processes) predicts behavioral performance is compelling.

Major Weaknesses and Limitations

(1) The central claim of 1 Hz rhythmic sampling is insufficiently validated. The windowing procedure (0.5s windows with 0.25s step) inherently restricts frequency resolution, potentially biasing toward low-frequency components like 1 Hz. Testing different window durations or providing controls would significantly strengthen this claim.

(2) The study lacks a baseline or control condition without distractors. This makes it difficult to determine whether the distractor-related decoding signals or the 1 Hz effect reflect genuine distractor processing or more general task dynamics.

(3) The pairwise decoding accuracies for distractor categories hover close to chance (~55%), raising concerns about robustness. While statistically above chance, the small effect sizes need careful interpretation, particularly when linked to behavior.

(4) Neither target nor distractor signal strength (SSVEP amplitude) correlates with behavioral accuracy. The study instead relies heavily on relative phase, which-while interesting-may benefit from additional converging evidence.

(5) Phase analysis is performed between different types of signals hindering their interpretability (time-resolved SSVEP amplitude and time-resolved decoding accuracy).

The authors largely achieved their stated goal of assessing rhythmic sampling of distractors. However, the conclusions drawn - particularly regarding the presence of 1 Hz rhythmicity - rest on analytical choices that should be scrutinized further. While the observed phase-performance relationship is interesting and potentially impactful, the lack of stronger and convergent evidence on the frequency component itself reduces confidence in the broader conclusions.

If validated, the findings will advance our understanding of attentional dynamics and competition in complex visual environments. Demonstrating that ignored distractors can be rhythmically sampled at similar frequencies to targets has implications for models of attention and cognitive control. However, the methodological limitations currently constrain the paper's impact.

Additional Considerations

• The use of EEG-fMRI is mentioned but not leveraged. If BOLD data were collected, even exploratory fMRI analyses (e.g., distractor modulation in visual cortex) could provide valuable converging evidence.

• In turn, removal of fMRI artifacts might introduce biases or alter the data. For instance, the authors might consider investigating potential fMRI artifact harmonics around 1 Hz to address concerns regarding induced spectral components.

Comments on revisions:

The authors have addressed my previous points, and the manuscript is substantially improved. The key methodological clarifications have been incorporated, and the interpretation of findings has been appropriately moderated. I have no further major concerns.

---

## [Author Response]

The following is the authors’ response to the original reviews

**Reviewer 1:**
(1) In general, the representation of target and distractor processing is a bit of a reach. Target processing is represented by SSVEP amplitude, which is most likely going to be related to the contrast of the dots, as opposed to representing coherent motion energy, which is the actual target. These may well be linked (e.g., greater attention to the coherent motion task might increase SSVEP amplitude), but I would call it a limitation of the interpretation. Decoding accuracy of emotional content makes sense as a measure of distractor processing, and the supplementary analysis comparing target SSVEP amplitude to distractor decoding accuracy is duly noted.

We agree with the reviewer. The SSVEP amplitude of the target at the whole trial level indeed reflected the combined effect of the stimulus parameters (e.g., contrast of the moving dots) as well as attention. However, the time course of the target SSVEP amplitude within a trial, derived from the moving window analysis, reflected the temporal fluctuations of target processing, since the stimulus parameters remained the same during the trial. We now make this clearer in the revised manuscript.

(2) Comparing SSVEP amplitude to emotional category decoding accuracy feels a bit like comparing apples with oranges. They have different units and scales and probably reflect different neural processes. Is the result the authors find not a little surprising in this context? This relationship does predict performance and is thus intriguing, but I think this methodological aspect needs to be discussed further. For example, is the phase relationship with behaviour a result of a complex interaction between different levels of processing (fundamental contrast vs higher order emotional processing)?

Traditionally, the SSVEP amplitude at the distractor frequency is used to quantify distractor processing. Given that the target SSVEP amplitude is stronger than that of the distractor, it is possible that the distractor SSVEP amplitude is contaminated by the target SSVEP amplitude due to spectral power leakage; see Figure S4 for a demonstration of this. Because of this issue we therefore introduced the use of decoding accuracy as an index of distractor processing. The lack of correlation between the distractor SSVEP amplitude and the distractor decoding accuracy, although it is kind of like comparing apples with oranges as pointed out by the reviewer, serves the purpose of showing that these two measures are not co-varying, and the use of decoding accuracy is free from the influence of the distractor SSVEP amplitude which is influenced by the target SSVEP amplitude. Also, to address the apples-vs-oranges issue, the correlation was computed on normalized time series, in which a z-score time series replaced the original time series so that the correlated variables are dimensionless. Regarding the question of assessing the relation between behavior and different levels of processing, we do not have means to address it, given that we are not able to empirically separate the effects of stimulus parameters versus attention.

**Reviewer 2:**
(1) Incomplete Evidence for Rhythmicity at 1 Hz: The central claim of 1 Hz rhythmic sampling is insufficiently validated. The windowing procedure (0.5s windows with 0.25s step) inherently restricts frequency resolution, potentially biasing toward low-frequency components like 1 Hz. Testing different window durations or providing controls would significantly strengthen this claim.

We appreciate the reviewer’s insightful suggestion. In response, we tested different windowing parameters, e.g., 0.1s sliding window with a 0.05s step size. Figure S5 demonstrates that the strength of both target and distractor processing fluctuates around ~1 Hz, both at the individual and group levels. Additionally, Figures S6(A) and S6(B) show that the relative phase between target and distractor processing time series exhibits a uniform distribution across subjects. In terms of the relation between relative phase and behavior, Figure S6(C) illustrates two representative cases: a high-performing subject with 84.34% task accuracy exhibited a relative phase of 0.9483π (closer to π), while a low-performing subject with 30.95% accuracy showed a phase of 0.29π close to 0. At the group level, a significant positive correlation between relative phase and task performance was found (r = 0.6343, p = 0.0004), as shown in Figure S6(D). All these results, aligning closely with our original findings (0.5s window length and 0.25s step size), suggest that the conclusions are not dependent on windowing parameters. We discuss these results in the revised manuscript.

To further validate our findings, we also employed the Hilbert transform to extract amplitude envelopes of the target and distractor signals on a time-point-by-time-point basis, providing a window-free estimate of signal strength (Figures R3 and R4). The results remain consistent with both the original findings and the new sliding window analyses (Figure S6). Specifically, Figure S7 reveals ~1 Hz fluctuations in target and distractor processing at both individual and group levels. Figures S8(A) and S8(B) confirm a uniform distribution of the relative phase across subjects. In Figure S8(C), the relative phase was 0.9567π for a high-performing subject (84.34% accuracy) and 0.2247π for a low-performing subject (28.57% accuracy). At the group level, a significant positive correlation was again observed between relative phase and task performance (r = 0.4020, p = 0.0376), as shown in Figure S8(D).

(2) No-Distractor Control Condition: The study lacks a baseline or control condition without distractors. This makes it difficult to determine whether the distractor-related decoding signals or the 1 Hz effect reflect genuine distractor processing or more general task dynamics.

The lack of a no-distractor control condition is certainly a limitation and will be acknowledged as such in the revised manuscript. However, given that our decoding results are between two different classes of distractors, we are confident that they reflect distractor processing.

(3) Decoding Near Chance Levels: The pairwise decoding accuracies for distractor categories hover close to chance (~55%), raising concerns about robustness. While statistically above chance, the small effect sizes need careful interpretation, particularly when linked to behavior.

This is an important point. To test robustness, we have implemented a random permutation procedure in which trial labels were randomly shuffled to construct a nullhypothesis distribution for decoding accuracy. We then compared the decoding accuracy from the actual data to this distribution. Figure S9 shows the results based on 1,000 permutations. For each of the three pairwise classifications—pleasant vs. neutral, unpleasant vs. neutral, and pleasant vs. unpleasant—as well as the three-way classification, the actual decoding accuracies fall far outside the null-hypothesis distribution (p < 0.001), and the effect size in all four cases is extremely large. These findings indicate that the observed decoding accuracies are statistically significant and robust in terms of both statistical inference and effect size.

(4) No Clear Correlation Between SSVEP and Behavior: Neither target nor distractor signal strength (SSVEP amplitude) correlates with behavioral accuracy. The study instead relies heavily on relative phase, which - while interesting - may benefit from additional converging evidence.

We felt that what the reviewer pointed out is actually the main point of our study, namely, it is not the target or distractor strength over the whole trial that matters for behavior, it is their temporal relationship within the trial that matters for behavior. This reveals a novel neuroscience principle that has not been reported in the past. We have stressed this point further in the revised manuscript.

(5) Phase-analysis: phase analysis is performed between different types of signals hindering their interpretability (time-resolved SSVEP amplitude and time-resolved decoding accuracy).

The time-resolved SSVEP amplitude is used to index the temporal dynamics of target processing whereas the time-resolved decoding accuracy is used to index the temporal dynamics of distractor processing. As such, they can be compared, using relative phase for example, to examine how temporal relations between the two types of processes impact behavior. This said, we do recognize the reviewer’s concern that these two processes are indexed by two different types of signals. We thus normalized each time course using zscoring, making them dimensionless, and then computed the temporal relations between them.

Appraisal of Aims and Conclusions:The authors largely achieved their stated goal of assessing rhythmic sampling of distractors. However, the conclusions drawn - particularly regarding the presence of 1 Hz rhythmicity - rest on analytical choices that should be scrutinized further. While the observed phaseperformance relationship is interesting and potentially impactful, the lack of stronger and convergent evidence on the frequency component itself reduces confidence in the broader conclusions.Impact and Utility to the Field:If validated, the findings will advance our understanding of attentional dynamics and competition in complex visual environments. Demonstrating that ignored distractors can be rhythmically sampled at similar frequencies to targets has implications for models of attention and cognitive control. However, the methodological limitations currently constrain the paper's impact.

Thanks for these comments and positive assessment of our work’s potential implications and impact. As indicated above, in the revision process, we have carried out a number of additional analyses, some suggested by the reviewers, and the results of the additional analyses, now included in the Supplementary Materials, served to further validate the main findings and strengthen our conclusions.

Additional Context and Considerations:(1) The use of EEG-fMRI is mentioned but not leveraged. If BOLD data were collected, even exploratory fMRI analyses (e.g., distractor modulation in visual cortex) could provide valuable converging evidence.

Indeed, leveraging fMRI data in EEG studies would be very beneficial, as has been demonstrated in our previous work. However, given that this study concerns the temporal relationship between target and distractor processing, it is felt that fMRI data, which is known to possess low temporal resolution, has limited potential to contribute. We will be exploring this rich dataset in other ways in the future, where we will be integrating the two modalities for more insights that are not possible with either modality used alone.

**Author response image 1. sa3fig1:** Appyling moving window analysis (0. 02s window duration and 0.01 step size) to a different EEG-fMRI dataset. (**A**) The amplitude time series of the 4.29 Hz component and the Fourier spectrum. (**B**) The group level Fourier spectrum. At both individual and group level, no 1 Hz modulation is observed, suggesting that the 1 Hz modulation observed in our data is not introduced by the artifact removal procedure.

(2) In turn, removal of fMRI artifacts might introduce biases or alter the data. For instance, the authors might consider investigating potential fMRI artifact harmonics around 1 Hz to address concerns regarding induced spectral components.

We have done extensive work in the area of simultaneous EEG-fMRI and have not encountered artifacts with a 1Hz rhythmicity. Our scanner artifact removal procedure is very standardized. As such, it stands to reason that if the 1Hz rhythmicity observed here results from the artifact removal process, it should also be present in other datasets where the same preprocessing steps were implemented. We tested this using another EEG-fMRI dataset (Rajan et al., 2019) . Author response image 1 shows that the EEG power time series of the new dataset doesn't have 1 Hz rhythmicity, whether at the individual level or at the group level, suggesting that the 1 Hz rhythmicity reported in the manuscript is not coming from the removal of the scanner artifacts, but instead reflects true rhythmic sampling of stimulus information. Also, the fact that the temporal relations between target processing and distractor processing at 1Hz impact behavior is another indication that the 1Hz rhythmicity is a neuroscientific effect, not an artifact.

References

Rajan, A., Siegel, S. N., Liu, Y., Bengson, J., Mangun, G. R., & Ding, M. (2019). Theta Oscillations Index Frontal Decision-Making and Mediate Reciprocal Frontal–Parietal Interactions in Willed Attention. Cerebral Cortex, 29(7), 2832–2843. https://doi.org/10.1093/cercor/bhy149